# SparseLoRA: Accelerating LLM Fine-Tuning with Contextual Sparsity

Samir Khaki [*1] Xiuyu Li [*†2] Junxian Guo [*3] Ligeng Zhu [3] Konstantinos N. Plataniotis [1]
Amir Yazdanbakhsh [4] Kurt Keutzer [2] Song Han [3] Zhijian Liu [3]
https://z-lab.ai/projects/sparselora

## Abstract

Fine-tuning LLMs is both computationally and memory-intensive. While parameter-efficient fine-tuning methods, such as QLoRA and DoRA, reduce the number of trainable parameters and lower memory usage, they do not decrease computational cost. In some cases, they may even slow down fine-tuning. In this paper, we introduce *SparseLoRA*, a method that accelerates LLM fine-tuning through contextual sparsity. We propose a lightweight, training-free *SVD sparsity estimator* that dynamically selects a sparse subset of weights for loss and gradient computation. Also, we systematically analyze and address sensitivity across layers, tokens, and training steps. Our experimental results show that SparseLoRA reduces computational cost by up to **2.2×** and a measured speedup of up to **1.6×** while maintaining accuracy across various downstream tasks, including commonsense and arithmetic reasoning, code generation, and instruction following.

## 1. Introduction

Large language models (LLMs) are trained on vast, general-domain datasets. They are often fine-tuned to improve their performance in specific domains (Saab et al., 2024) or to align their predictions with user preferences (Zhang et al., 2024a). However, fine-tuning very large models can be prohibitively expensive, both in terms of memory requirements and computational costs.

Extensive efforts have been made in parameter-efficient fine-tuning (PEFT) to reduce memory consumption of LLM fine-tuning. LoRA (Hu et al., 2022) represents weight updates using low-rank approximations. Building upon this, many

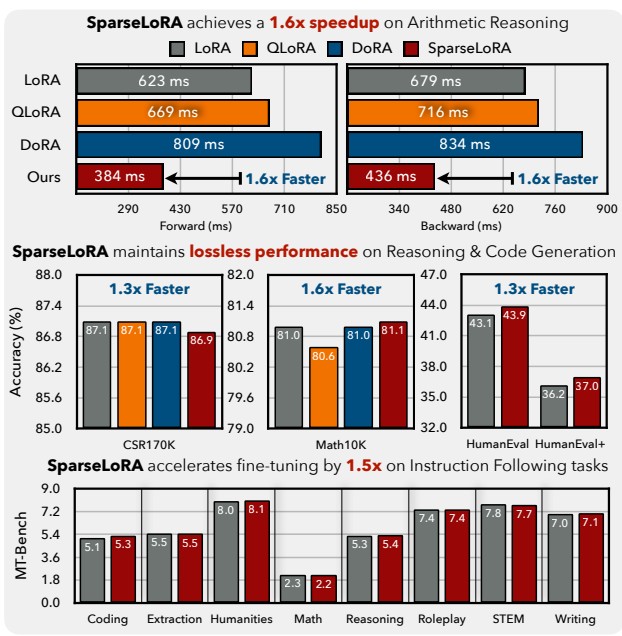

Figure 1: Many recent parameter-efficient fine-tuning methods, such as QLoRA and DoRA, do not reduce compute. Our SparseLoRA accelerates LLM fine-tuning with minimal accuracy loss across a range of downstream tasks, including commonsense reasoning, math reasoning, code generation, and instruction following. See Section 4 for more details.

follow-up methods (Dettmers et al., 2023; Liu et al., 2024c) have been proposed to further reduce the number of trainable parameters. While they are effective in reducing memory usage, they do not reduce computation. In fact, they can sometimes slow down fine-tuning due to the overhead they introduce: DoRA is 20% slower than LoRA (see Figure 1).

In this paper, we present **SparseLoRA** to accelerate LLM fine-tuning with contextual sparsity, making it both memory- and computation-efficient. Contextual sparsity has already been used in accelerating LLM inference (Liu et al., 2023b). SparseLoRA shows for the first time that it can also play a role in LLM fine-tuning, where (1) only a sparse subset of weights is required for loss and gradient computation, and (2) this sparse subset needs to be determined based on the input sequence or tokens. To realize this, we propose

---

[*]Equal contribution [†]Project lead [1]University of Toronto [2]UC Berkeley [3]MIT [4]Google DeepMind. Correspondence to: Samir Khaki <samir.khaki@mail.utoronto.ca>.

*Proceedings of the 42nd International Conference on Machine Learning*, Vancouver, Canada. PMLR 267, 2025. Copyright 2025 by the author(s).

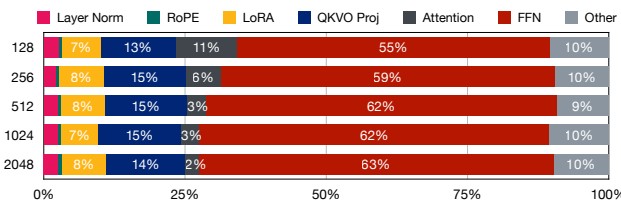

Figure 2: Runtime breakdown of LLaMA3-8B fine-tuning under different sequence lengths.

using an **SVD sparsity estimator** to identify which channels should be activated based on samples within each batch. It is very *lightweight*, adding only 0.05% FLOPs and 0.8% runtime overhead to fine-tuning. It is *training-free*, unlike the look-ahead predictor proposed in Deja Vu (Liu et al., 2023b), which leads to better generalization across datasets.

We have conducted systematic sensitivity analysis across multiple dimensions. First, each layer responds differently to sparsity, so we apply non-uniform sparsity based on layer sensitivity analysis. Second, output tokens are much more sensitive to pruning than context tokens, so we apply sparsity only to context tokens. Finally, early iterations in fine-tuning are more sensitive, so we run dense fine-tuning in the early iterations and switch to sparse fine-tuning for the remainder.

Evaluated across a diverse set of benchmarks, SparseLoRA achieves a computational cost reduction of up to 2.2× and a wall-clock speedup of up to 1.6× while maintaining accuracy on various downstream tasks, including commonsense and arithmetic reasoning, code-generation, and complex instruction following. To the best of our knowledge, this is the first work to leverage contextual sparsity for accelerating LLM fine-tuning. We believe that our work will inspire future research into fine-tuning methods that optimize both parameter and computational efficiency.

## 2. Related Work

**Contextual Sparsity in LLMs.** Sparsity has long been employed to enhance the efficiency of neural networks (Han et al., 2016b; 2015; 2016a; Frantar & Alistarh, 2023; Sun et al., 2024). Recently, there has been research on a more dynamic approach: accelerating LLMs inference by leveraging contextual sparsity (Liu et al., 2023b; Song et al., 2023; Xue et al., 2024; Alizadeh et al., 2024; Lee et al., 2024; Akhauri et al., 2024; Liu et al., 2024a; Zhang et al., 2025b) at test time. Unlike static sparsity, contextual sparsity is a phenomenon where significant portions of a model's hidden states dynamically contain zero-valued neurons based on the input context. This allows for input-dependent sparse computation without compromising outcomes. While such activation sparsity naturally emerges in ReLU-based FFNs (Li et al., 2023b; Mirzadeh et al., 2023; Liu et al., 2023b; Alizadeh et al., 2024), newer architectures often employ non-ReLU activations (Team, 2023; 2024c;a;b) that create dif-

ferent sparsity patterns, necessitating alternative methods to reintroduce and exploit sparsity.

Recent work has explored various techniques, including continued pretraining (Song et al., 2023; Zheng et al., 2024a; Zhang et al., 2024c; Song et al., 2024a;b; Xue et al., 2024) and magnitude pruning with specific metrics (Lee et al., 2024; Akhauri et al., 2024; Liu et al., 2024a; Zhang et al., 2025b), to leverage sparsity in LLMs. These approaches aim to reduce memory usage and computation time, with some work deploying small neural networks to predict non-zero activations (Liu et al., 2023b; Alizadeh et al., 2024; Akhauri et al., 2024; Song et al., 2023; Xue et al., 2024). In this work, we propose a novel approach to extend the benefits of contextual sparsity to the fine-tuning process for the first time, which addresses the associated challenges and accelerates fine-tuning without compromising performance.

**Memory-Efficient Fine-tuning.** As language models grow larger, memory-efficient fine-tuning methods have become crucial. Parameter-efficient fine-tuning (PEFT) techniques address this challenge by updating only a small subset of parameters. LoRA (Hu et al., 2022) employs low-rank matrices to adjust pretrained model weights, sparking a rich line of research with numerous works proposing improvements and variations (Dettmers et al., 2023; Shi et al., 2023; Qiu et al., 2023; Chen et al., 2024; Kopiczko et al., 2024; Liu et al., 2024c; Meng et al., 2024; Hayou et al., 2024; Wang et al., 2024; Wang & Liang, 2024; Liu et al., 2024d; Pan et al., 2024; Yang et al., 2024). Among these, DoRA (Liu et al., 2024c) reparameterizes weight matrices to achieve more effective optimization. QLoRA (Dettmers et al., 2023) combines quantization with low-rank adapters, enabling fine-tuning of large models on a single GPU.

Recent research has also focused on exploiting the low-rank structure of PEFT (Nikdan et al., 2024; Agarwal et al., 2024; Zhang et al., 2025a). GaLore (Zhao et al., 2024) and its weight-quantized variant (Zhang et al., 2024b) leverage the low-rank property of weight gradients to reduce optimizer state memory, while WeLore (Jaiswal et al., 2024) investigates how low-rank weights emerge from low-rank gradients during training. While these advancements make LLM adaptation more memory-efficient and accessible, they primarily focus on reducing memory usage or improving the accuracy, sometimes even at the cost of increased computation time. Our approach addresses this missing piece by focusing on compute efficiency, complementing existing memory-efficient techniques to enable truly resource-efficient fine-tuning.

**Computation-Efficient Training.** Prior work has explored sparsity to accelerate LLM training through various approaches (Thangarasa et al., 2023; Chen et al., 2024; Mozaffari et al., 2024). LongLoRA (Chen et al.,

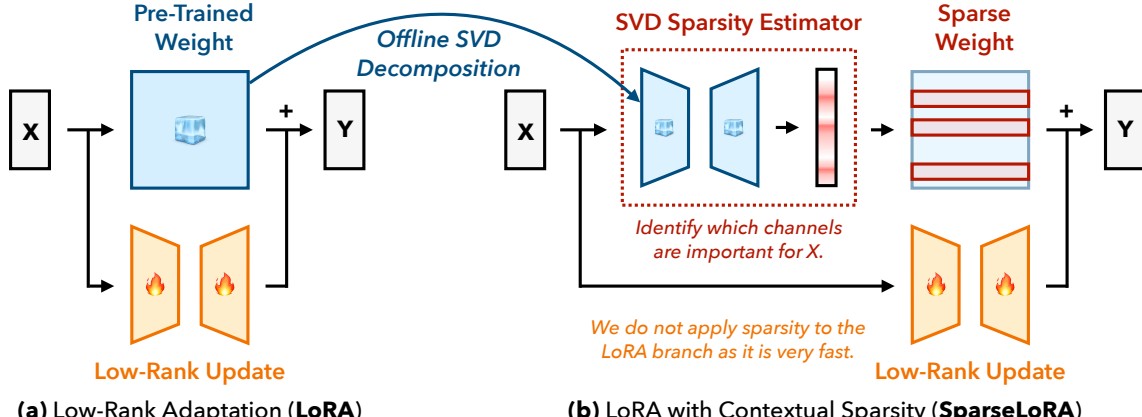

**(a)** Low-Rank Adaptation (**LoRA**)  **(b)** LoRA with Contextual Sparsity (**SparseLoRA**)

Figure 3: SparseLoRA accelerates LLM fine-tuning with contextual sparsity. It first performs an offline SVD decomposition of the pre-trained weights to construct the SVD sparsity estimator. During fine-tuning, it relies on the SVD sparsity estimator to identify which weight channels are needed for the given input sequence. It then sparsely computes by slicing the weights on the fly. Note that sparsity is applied only to the main branch, as the LoRA branch is typically very fast.

2024) enables efficient context window extension via shifted sparse attention during fine-tuning while retaining standard attention at inference, reducing quadratic computation costs. General sparse training techniques have demonstrated promising FLOPs reduction through combinations of sparse pre-training and further fine-tuning (Thangarasa et al., 2023; Mozaffari et al., 2024). Other methods employ dynamic sparsity patterns through gradient-based selection (Zhou et al., 2021; Li et al., 2022). Yet these techniques either focus on long-context scenarios where attention dominates computation, or rely on unstructured sparsity, whose theoretical gains are challenging to realize on consumer-grade GPUs. While there have been recent attempts to apply structural sparsity (Ma et al., 2024; Chen et al., 2025), they either are not memory efficient or only partially accelerate fine-tuning (e.g., the backward pass), limiting their effectiveness. Our work introduces structured contextual sparsity with training-free dynamic selection, enabling practical end-to-end acceleration while remaining memory-efficient.

## 3. Method

In this section, we present the design of our SparseLoRA. As shown in Figure 2, linear layers dominate LoRA fine-tuning runtime in conventional settings. Therefore, we apply **dynamic channel sparsity** to the main branch of LoRA fine-tuning while keeping the LoRA branch dense. This approach selectively activates only the most important neurons in FFN and attention layers. Since the main branch accounts for the vast majority of computation, and the sparsity introduced is structural and hardware-friendly, we achieve significant efficiency improvements without altering which parameters are updated. By sparsifying only the base model while keeping LoRA intact, SparseLoRA maintains both

**memory- and computation-efficient** fine-tuning with no impact on inference performance.

### 3.1. Sparse Neuron Selection Criteria

To achieve compute-efficient fine-tuning while maintaining effectiveness, we require dynamic rather than static sparsity patterns that adapt to each input. Prior research has explored contextual activation sparsity in LLM inference (Liu et al., 2023b; Alizadeh et al., 2024; Akhauri et al., 2024; Liu et al., 2024a). However, these *inference* methods target single-token computations in auto-regressive generation and do not directly translate to *fine-tuning*, which has distinct workloads that consist of multiple sequences of tokens in a batch. To bridge this gap, we first define **"oracle"** criteria for fine-tuning neuron selection using intermediate activations, establishing an ideal but computationally infeasible metric. This oracle then guides the practical development of efficient approximations that enable on-the-fly channel selection for sparse computation. We categorize the linear layers in LLMs into three types – FFN, VO projections, and QK projections – and propose two oracle criteria tailored to their properties as follows.

#### 3.1.1. SELECTION WITH L2 NORM

Motivated by the extreme sparsity in input activations for certain linear layers, such as the SiLU-induced sparsity in the down projection of FFNs (Alizadeh et al., 2024; Lee et al., 2024; Song et al., 2023) as shown in Figure 4, we introduce an L2 Norm metric to identify and retain the most significant neurons as the first oracle criterion. Importantly, this approach allows us to naturally extend the sparsity pattern to the preceding linear layers – specifically using FFN as an example, the channel indices selected for the down

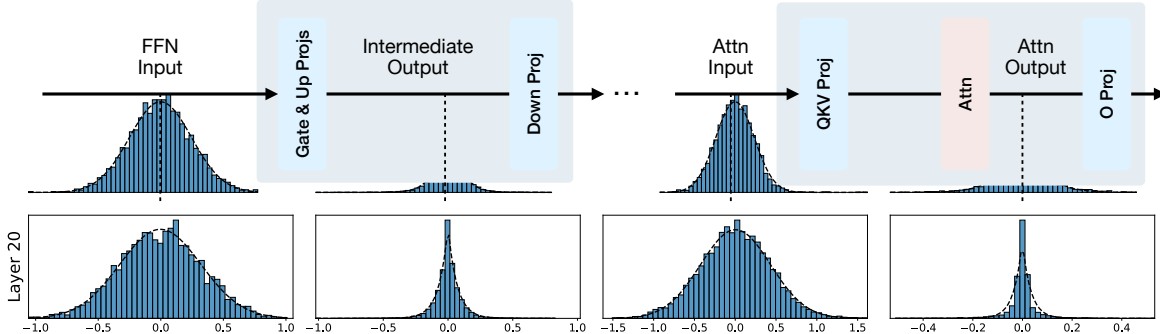

Figure 4: Activation value distributions across all tokens of a sequence for layer 20 of LLaMA2-7B. Each column shows inputs to: $W_{gate,up}$, $W_{down}$ of FFNs, and $W_{Q,K,V}$, $W_O$ of attention projections. The inputs to $W_{down}$ and $W_O$ follow Laplace distributions, enabling higher sparsity when using the L2 norm metric. $W_{gate,up}$ is pruned alongside corresponding $W_{down}$ channels and $W_V$ is pruned alongside corresponding $W_O$ channels, while $W_{Q,K}$ requires a different criterion.

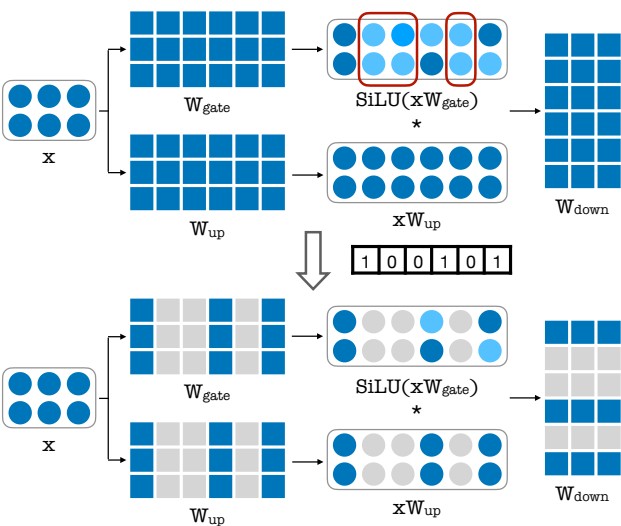

Figure 5: An illustration of how sparsity is applied to FFNs.

projection can be applied to both up and gate projections, as illustrated in Figure 5.

While prior work has applied similar magnitude-based metrics for inference with single-token (Liu et al., 2023b; Lee et al., 2024; Akhauri et al., 2024; Liu et al., 2024a), we extend this to fine-tuning by considering cumulative activations across all samples and tokens in a batch. Despite recent exploration of incorporating gradient and weight information (Akhauri et al., 2024; Sun et al., 2024), we find the L2 norm of activations alone provides a simple yet effective criterion, as activations generally contain more influential outliers than weights (Xiao et al., 2023). We apply this metric to both FFNs and the outer channels of value and output (VO) projections in attention, where input activations to output projections show similar sparsity characteristics as demonstrated in Figure 4.

### 3.1.2. SELECTION WITH QK NORM

For attention blocks, we aim to sparsify the linear layers as well: query, key, value, and output projections. While the L2 Norm metric works well for VO projections, it proves impractical for QK projections due to their lower input activation sparsity as shown in Figure 4. Previous approaches exploring contextual sparsity during inference have proposed identifying and pruning unimportant attention heads (Liu et al., 2023b; Akhauri et al., 2024). However, this head-level pruning strategy proves problematic for fine-tuning scenarios that process multiple tokens simultaneously. Unlike FFNs where we can selectively choose from 11,384 channels in LLaMA2-7B, pruning entire attention heads (e.g., removing 1 out of 32) significantly constrains our pruning granularity and risks losing critical information. We empirically verify in § 4.3 that this coarse-grained approach leads to degraded performance. Additionally, a detailed analysis of attention head activation patterns during fine-tuning is provided in Appendix A.1.

To address this challenge, we introduce an oracle criterion for sparsifying QK projections based on attention scores, targeting channels with minimal contributions. Specifically, we define a proxy metric that quantifies each channel's importance and sparsify those with the lowest values. Given query and key projections $\mathbf{Q}, \mathbf{K} \in \mathbb{R}^{(B \times L) \times D}$, where $B$ is the batch size, $L$ is the sequence length, and $D$ is the hidden dimension, we compute their L2 norms across the flattened batch and sequence dimensions:

$$\mathbf{q} = \|\mathbf{Q}\|_2, \quad \mathbf{k} = \|\mathbf{K}\|_2.$$

The element-wise product of these normalized scores serves as our importance metric:

$$\mathbf{s} = \mathbf{q} \odot \mathbf{k}.$$

We retain the top $n$ channels based on $\mathbf{s}$, where $n$ is determined by the desired sparsity ratio.

This approach ensures that only the projection channels contributing most significantly to the attention scores are retained. Compared to the L2 norm criterion from FFNs, this method better preserves the original computational outcomes. As demonstrated in Figure 6, our proposed oracle criterion maintains an attention map much more similar to the original dense QK computation compared to those derived by L2 norm or random pruning.

## 3.2. SVD Sparsity Estimator

The oracle criteria demonstrate that adaptively sparsifying the backbone weights during fine-tuning can significantly reduce computation while maintaining model quality. However, computing these oracle patterns requires partial dense computation such as the gate and up projections for FFN intermediate activations and QK projections for QK norms, making it impractical and negating potential speedup benefits. The utility of SVD in identifying the most significant components of weight matrices is well-established in model compression (Nikdan et al., 2024; Li* et al., 2025; Liu et al., 2024b; Zhang et al., 2025b), suggesting its potential in our setting. To address this challenge, we introduce an efficient SVD-based low-rank sparsity estimator that dynamically selects channels with minimal overhead. Our method directly approximates the oracle criterion using top-k singular value decomposition (SVD) of the model weights, in contrast to prior approaches that employ learned low-rank predictors (Liu et al., 2023b; Alizadeh et al., 2024; Akhauri et al., 2024), which raise concerns about generalization across different datasets and tasks.

---

**Algorithm 1** SVD Sparsity Estimator

---

```
# Assume input tensor x of shape (B, S, D1)
# Weight W of shape (B, D1, D2), SVD rank of k

- # Compute activations with oracle
- out = torch.bmm(x, W)

+ # Compute low-rank SVD weights (saved offline)
+ U, S, V = torch.linalg.svd(W, full_matrices=False)
+ W_A = U[:, :k] @ torch.diag(S[:k]).sqrt()
+ W_B = torch.diag(S[:k]).sqrt() @ V[:k, :]

+ # Compute activations with loaded SVD estimator
+ out = torch.bmm(torch.bmm(x, W_A), W_B)

# Obtain channel indices with corresponding metric
indices = metric(out)
```

---

The core idea is straightforward: instead of training a predictor to map inputs to sparsity masks, we project inputs onto a low-rank SVD decomposition of the original weights and compute the oracle metric from these projected activations, as detailed in Algorithm 1. This approach produces sparsity masks that closely match those obtained from

the full model while maintaining efficiency. The SVD decomposed components are computed offline and loaded at the start of fine-tuning alongside the model weights. Notably, while low-rank module overheads can be significant in inference-time sparsity methods due to memory-bound execution (Akhauri et al., 2024; Liu et al., 2023b), our approach introduces minimal overhead (less than 1% of runtime) since the lightweight SVD projections are negligible compared to the matrix multiplications in the compute-bound LLM fine-tuning. This enables us to achieve dynamic sparsity with negligible computational overhead, preserving both fine-tuning efficiency and model performance.

## 3.3. Sensitivity Analysis

**Layer Sensitivity: Adaptive Sparsity Configuration** The inherent contextual sparsity across layers in LLMs often varies significantly (Liu et al., 2023b; 2024a). Moreover, the importance of individual layers and their contributions to fine-tuning can differ substantially (Gromov et al., 2024; Zheng et al., 2024a), necessitating layer-specific sparsity configurations for optimal performance. To determine these configurations, we conduct a systematic layer sensitivity analysis using a subset of the Commonsense Reasoning task proposed by Hu et al. (2023) as a proxy. Our analysis evaluates how different sparsity ratios affect each layer's performance independently – starting with a densely fine-tuned model, we progressively increase the sparsity ratio for each layer while keeping others dense, measuring performance each time to generate layer-specific sensitivity curves.

The results for LLaMA2-7B, shown in Figure 7, reveal that deeper layers contain more redundant information and are more amenable to sparsification than shallower layers, aligning with inference-time observations from Gromov et al. (2024). These sensitivity metrics enable us to apply aggressive sparsity to deeper, resilient layers while maintaining shallower ones dense, optimizing the performance-efficiency trade-off during fine-tuning.

**Token Sensitivity: Context-Output Aware Sparsity** The effectiveness of sparsity varies not only across layers but also across tokens within a sequence. In LLM fine-tuning, input sequences typically consist of a *context* (the prefix tokens provided as input) and *output* tokens (the target tokens used for loss computation). We find that applying uniform sparsity across all tokens degrades performance, as output tokens play a more critical role in optimization.

To address this, we propose a context-output aware sparsity strategy, selectively preserving dense computation for output tokens while applying sparsity to the context. This ensures that fine-tuning retains full expressiveness where it matters most while still benefiting from reduced computation. Unlike heuristic-based token importance sampling, our approach exploits a natural structural distinction – context

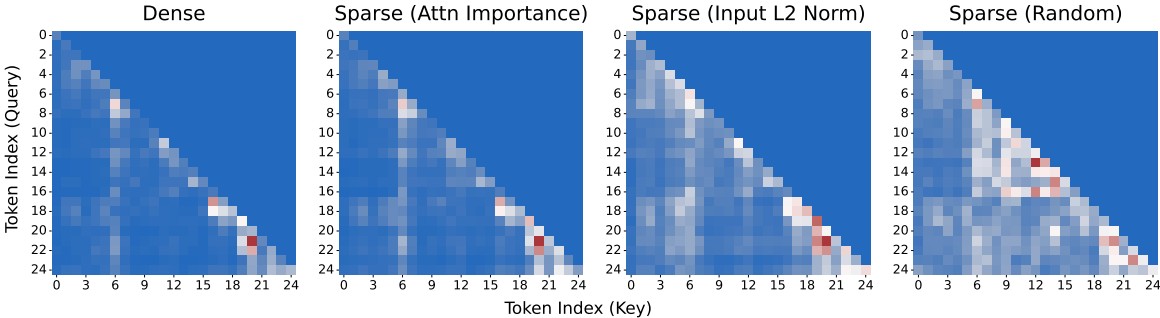

Figure 6: Comparison of attention maps for different pruning strategies at 50% sparsity. The figure shows the attention maps for the last 25 tokens of head 7 using dense full attention, pruning based on attention importance (ours), pruning based on input L2 norm (simlar to FFNs), pruning randomly.

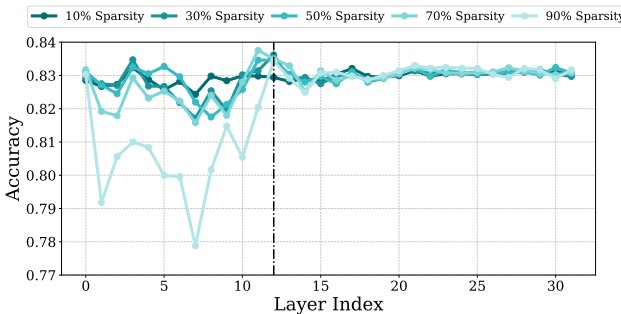

Figure 7: Sensitivity analysis on layer-wise sparsity of LLaMA2-7B.

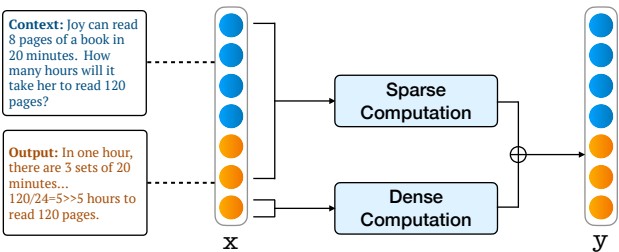

Figure 8: Output tokens go through the dense computation in our context-output aware sparsity strategy. The final outputs are gathered from both sparse and dense results.

tokens are inherently less sensitive to precise weight updates than output tokens, as illustrated in Figure 8. This strategy significantly mitigates the reconstruction errors between sparse and dense training, particularly in early fine-tuning steps where maintaining gradient signal is crucial. This simple yet effective approach improves computational efficiency while preserving fine-tuning performance.

**Step Sensitivity: Progressive Sparse Fine-tuning** To balance efficiency and model quality, we incorporate a hybrid approach in our fine-tuning process. Recent studies suggest that when doing incorporating sparsity in training,

maintaining dense computations for a small portion of steps can significantly enhance final convergence (Lu et al., 2023; Thangarasa et al., 2023; Bambhaniya et al., 2024) with minimal impact on overall speed-up. In our SparseLoRA design, we allow the initial steps, up to a maximum of 10% of the fine-tuning process, to remain dense. This approach ensures the model establishes a strong foundation early on while still benefiting from sparse training's efficiency in later stages. A detailed analysis of this hybrid approach's impact on performance and efficiency can be found in § 4.3.

## 4. Experiments

### 4.1. Setup

**Benchmarks.** We conduct experiments on five downstream tasks. The first set focuses on commonsense reasoning (referred to as CSR170K) and includes eight datasets: BoolQ (Clark et al., 2019), PIQA (Bisk et al., 2020), SIQA (Sap et al., 2019), HellaSwag (Zellers et al., 2019), WinoGrande (Sakaguchi et al., 2021), ARC-Easy and ARC-Challenge (Clark et al., 2018), and OpenbookQA (Mihaylov et al., 2018). The second set focuses on arithmetic reasoning (referred to as Math10K) and includes three benchmarks: GSM8K (Cobbe et al., 2021), MAWPS (Koncel-Kedziorski et al., 2016), and SVAMP (Patel et al., 2021)*. Following the practices established by Hu et al. (2023) and Liu et al. (2024c), we fine-tune our models on the combined training sets of all sub-tasks within each respective benchmark. We run each experiment five times, discard the highest and lowest performing runs, and report the average accuracy of the remaining three. We further assess the generality of our method on three additional tasks: sequence classification using the GLUE benchmark (Wang et al., 2018); instruction following, where we train on WizardLM (Xu et al., 2024)

---

*We exclude AQuA (Ling et al., 2017) since none of the methods in Hu et al. (2023) achieve better-than-random performance (i.e., significantly above 20% for a 5-choice multiple-choice task).

| | #FLOPs | Speedup | Average | BoolQ | PIQA | SIQA | HellaS | WinoG | ARC-e | ARC-c | OBQA |
|---|---|---|---|---|---|---|---|---|---|---|---|
| LLaMA2-7B | – | – | 31.4 | 51.0 | 49.5 | 32.4 | 25.0 | 20.4 | 23.7 | 22.1 | 26.6 |
| + LoRA | 100% | 1.0× | 82.3 | 70.7 | 84.8 | 81.4 | 90.0 | 85.8 | 87.6 | 74.3 | 84.7 |
| + QLoRA | 100% | 0.9× | 82.5 | 69.2 | 84.7 | 81.9 | 90.5 | 85.8 | 87.9 | 74.2 | 85.4 |
| + DoRA | 132% | 0.7× | 81.7 | 71.4 | 84.7 | 81.1 | 90.0 | 85.2 | 87.3 | 72.8 | 84.2 |
| **+ SparseLoRA** | 65% | 1.3× | 81.8 | 69.7 | 84.3 | 80.8 | 88.4 | 86.0 | 86.7 | 73.4 | 84.0 |
| LLaMA2-13B | – | – | 35.0 | 61.9 | 49.8 | 31.8 | 25.6 | 17.2 | 33.3 | 30.5 | 29.6 |
| + LoRA | 100% | 1.0× | 84.7 | 72.0 | 86.7 | 82.2 | 91.0 | 89.0 | 90.9 | 80.5 | 85.6 |
| **+ SparseLoRA** | 61% | 1.3× | 85.0 | 74.1 | 87.1 | 82.4 | 92.3 | 88.3 | 89.8 | 78.7 | 86.9 |
| LLaMA3-8B | – | – | 62.5 | 66.1 | 75.4 | 53.8 | 54.8 | 42.1 | 80.7 | 67.3 | 59.8 |
| + LoRA | 100% | 1.0× | 87.1 | 74.6 | 89.4 | 82.7 | 95.4 | 89.1 | 92.8 | 83.4 | 89.3 |
| + QLoRA | 100% | 0.9× | 87.1 | 74.3 | 89.3 | 83.1 | 95.3 | 88.7 | 92.9 | 83.8 | 89.4 |
| + DoRA | 132% | 0.8× | 87.1 | 74.5 | 89.4 | 83.0 | 95.4 | 88.8 | 93.2 | 84.0 | 88.9 |
| **+ SparseLoRA** | 65% | 1.3× | 86.9 | 75.0 | 89.6 | 82.8 | 94.9 | 88.7 | 92.7 | 82.9 | 88.3 |

Table 1: SparseLoRA delivers up to 1.3× speedup and reduces fine-tuning FLOPs by up to 39%, while maintaining performance comparable to existing methods such as LoRA, QLoRA and DoRA on commonsense reasoning benchmarks.

| | #FL. | Spd. | Avg. | GSM8K | SVAMP | MAWPS |
|---|---|---|---|---|---|---|
| LLaMA2-7B | – | – | 2.6 | 2.7 | 3.1 | 2.1 |
| + LoRA | 100% | 1.0× | 54.6 | 38.6 | 47.5 | 77.5 |
| + QLoRA | 100% | 0.9× | 55.0 | 36.2 | 49.7 | 79.1 |
| + DoRA | 132% | 0.7× | 54.5 | 38.4 | 48.4 | 77.5 |
| **+ SparseLoRA** | 73% | 1.2× | 53.7 | 37.6 | 46.4 | 77.9 |
| LLaMA2-13B | – | – | 13.4 | 4.9 | 18.8 | 16.4 |
| + LoRA | 100% | 1.0× | 63.5 | 50.2 | 59.1 | 81.0 |
| **+ SparseLoRA** | 70% | 1.3× | 62.7 | 49.5 | 57.1 | 81.5 |
| LLaMA3-8B | – | – | 33.5 | 25.0 | 38.4 | 37.0 |
| + LoRA | 100% | 1.0× | 81.0 | 71.8 | 80.3 | 90.9 |
| + QLoRA | 100% | 0.9× | 80.6 | 71.8 | 80.2 | 89.6 |
| + DoRA | 132% | 0.8× | 81.0 | 72.5 | 79.3 | 91.0 |
| **+ SparseLoRA** | 46% | 1.6× | 81.1 | 72.0 | 80.2 | 90.9 |

Table 2: SparseLoRA offers up to 1.6× speedup and reduces fine-tuning FLOPs by up to 54% on arithmetic reasoning tasks with accuracy comparable to existing methods.

and evaluate on MT-Bench (Zheng et al., 2023); and code generation, where we train on CodeFeedback (Chen et al., 2021) and test on HumanEval / HumanEval+ (Zheng et al., 2024b; Liu et al., 2023a).

**Models.** We use LLaMA2-7B/13B and LLaMA3-8B (Instruct) as our base models for fine-tuning. For instruction following and code generation benchmarks, we additionally use LLaMA3.1-8B.

**Baselines.** We compare our method with two PEFT methods, LoRA (Hu et al., 2022) and DoRA (Liu et al., 2024c).

While SparseLoRA is built on top of LoRA, it can, in principle, be applied to any other PEFT method. We include model training details in Table 12. All PeFT methods only fine-tune the QKVO projections using a rank of 32, a scaling factor $\alpha$ of 64, and no dropout. Efficiency metrics are derived from an NVIDIA A6000 GPU.

### 4.2. Main Results

Table 1 demonstrates results on the CSR170K benchmarks, demonstrating that SparseLoRA significantly reduces computational requirements while maintaining accuracy comparable to LoRA. For instance, on LLaMA2-13B, SparseLoRA achieves an average accuracy of 85.0, outperforming LoRA's 84.7, while requiring only 61% of its training cost with a 1.3× speedup in its training time. This trend extends to LLaMA3-8B, where SparseLoRA consistently reduces compute load while preserving competitive accuracy across tasks such as BoolQ, PIQA, SIQA, HellaSwag, WinoGrande, ARC, and OBQA.

Table 2 presents results on the Math10K benchmarks, further reinforcing SparseLoRA's compute–accuracy advantage. On LLaMA3-8B, SparseLoRA reduces training costs by 54% and achieves a 1.6× speedup on LoRA while maintaining strong performance on SVAMP and MAWPS while slightly surpassing LoRA on GSM8K. These findings demonstrate that structured contextual sparsity can significantly reduce computational overhead in parameter-efficient fine-tuning without sacrificing performance in commonsense and mathematical reasoning tasks.

**Natural Language Understanding.** We evaluate SparseLoRA using LLaMA3-8B on sequence classification

| | #FLOPs | Speedup | Average | COLA | STS-B | MRPC | RTE | SST2 | QNLI | WNLI | MNLI | QQP |
|---|---|---|---|---|---|---|---|---|---|---|---|---|
| LLaMA3-8B + LoRA | 100% | 1.0× | 87.3 | 65.8 | 88.8 | 87.7 | 82.8 | 96.4 | 95.7 | 45.5 | 91.8 | 89.8 |
| LLaMA3-8B + **SparseLoRA** | 61% | 1.3× | 87.7 | 66.3 | 89.3 | 88.6 | 82.9 | 96.6 | 96.6 | 55.9 | 91.8 | 89.8 |

Table 3: SparseLoRA accelerates fine-tuning for sequence classification on the GLUE benchmark by 1.3×.

| | #FLOPs | Speedup | Average | Coding | Extraction | Humanities | Math | Reasoning | Roleplay | STEM | Writing |
|---|---|---|---|---|---|---|---|---|---|---|---|
| LLaMA3.1-8B | – | – | 4.08 | 2.15 | 2.88 | 4.50 | 2.05 | 2.60 | 5.72 | 7.90 | 4.80 |
| + LoRA | 100% | 1.0× | 6.03 | 5.10 | 5.50 | 8.00 | 2.25 | 5.30 | 7.35 | 7.78 | 7.00 |
| + **SparseLoRA** | 53% | 1.5× | 6.06 | 5.30 | 5.45 | 8.10 | 2.20 | 5.35 | 7.35 | 7.70 | 7.05 |

Table 4: SparseLoRA delivers strong performance on instruction-following tasks in MT-Bench, achieving a 1.5× speedup while matching or exceeding LoRA across all categories.

with a subset of GLUE benchmark (Wang et al., 2018). Table 3 shows that SparseLoRA maintains competitive performance to the dense baseline with a 1.3× speedup.

**Instruction Following.** We evaluate SparseLoRA using LLaMA3.1-8B on the task of instruction following by fine-tuning on a subset of the WizardLM dataset (Xu et al., 2024) and reporting scores across eight tasks in the MT-Bench dataset (Zheng et al., 2023). We use GPT-4 to assess the quality of model responses. Table 4 shows that SparseLoRA maintains competitive performance to the dense baseline while achieving up to a 1.5× speedup.

**Code Generation.** We evaluate SparseLoRA using LLaMA2-7B and LLaMA3.1-8B on code generation by fine-tuning on a subset of the CodeFeedback dataset (Chen et al., 2021) and testing on the HumanEval benchmarks (Zheng et al., 2024b; Liu et al., 2023a). Table 5 shows that SparseLoRA maintains competitive performance to the dense baseline while achieving up to a 1.3× speedup.

| | #FLOPs | Speedup | HumanEval | HumanEval+ |
|---|---|---|---|---|
| LLaMA2-7B | – | – | 3.6 | 3.0 |
| + LoRA | 100% | 1.0× | 13.0 | 10.2 |
| + **SparseLoRA** | 73% | 1.2× | 12.8 | 11.0 |
| LLaMA3.1-8B | – | – | 30.9 | 27.9 |
| + LoRA | 100% | 1.0× | 43.1 | 36.2 |
| + **SparseLoRA** | 66% | 1.3× | 43.9 | 37.0 |

Table 5: SparseLoRA accelerates fine-tuning for code generation by up to 1.3× while maintaining performance.

**ARC-AGI.** We evaluate SparseLoRA using a fine-tuned version of LLaMA3.1-8B following Barc (Li et al., 2024) on test-time training for the abstract and reasoning corpus of artificial general intelligence (ARC-AGI) (Chollet, 2019). Table 6 shows that SparseLoRA maintains competitive per-

formance to the LoRA baseline on the pass@k (k=2,4) settings, while achieving up to a 1.3× speedup.

| | #FLOPs | Speedup | pass@2 | pass@4 |
|---|---|---|---|---|
| LLaMA3.1-8B | – | – | 24.2 | 29.0 |
| + LoRA | 100% | 1.0× | 33.7 | 37.4 |
| + **SparseLoRA** | 71% | 1.3× | 33.9 | 37.5 |

Table 6: SparseLoRA accelerates test-time-training on ARC-AGI by up to 1.3× while maintaining performance.

**Compatibility with PEFT Methods.** Techniques such as gradient checkpointing and quantization (*e.g.*, QLoRA (Dettmers et al., 2023), LoftQ (Li et al., 2023a)) primarily aim to reduce memory usage but often increase runtime, as shown in Figure 1. These methods are therefore orthogonal and complementary to our approach. As in Table 7, when combined with QLoRA, SparseLoRA achieves both lower memory usage and faster runtime.

### 4.3. Analysis

**SVD Sparsity Estimator.** The SVD sparsity estimator is key to SparseLoRA 's ability to apply contextual sparsity with minimal impact on performance. As shown in Table 8, our estimator achieves performance comparable to the Oracle method on the Math10K dataset, demonstrating its effectiveness. Using a rank 8 singular value decomposition of the base model weights, the SVD sparsity estimator is lightweight and training-free, adding only 0.8% overhead to the end-to-end runtime.

**Effect of Output Token Splitting.** We investigate the effectiveness of our context-output splitting strategy, which optimizes computational efficiency by preserving dense computation for output tokens while sparsifying the context tokens. To evaluate the impact of this design choice, we compare three configurations: (1) the baseline, which

|  | CSR170K | | | Math10K | | |
|---|---|---|---|---|---|---|
|  | #FL. | Spd. | Acc. | #FL. | Spd. | Acc. |
| LLaMA3-8B | – | – | 62.5 | – | – | 33.5 |
| + QLoRA | 100% | 1.0× | 87.1 | 100% | 1.0× | 80.6 |
| + **SparseQLoRA** | 65% | 1.2× | 86.9 | 60% | 1.3× | 80.8 |

Table 7: SparseLoRA can be combined with existing PeFT approaches, such as QLoRA, to accelerate fine-tuning while maintaining memory savings.

|  | #FLOPs | Runtime | Memory | Accuracy |
|---|---|---|---|---|
| Oracle | – | – | – | 81.4 |
| SVD | 0.05% | 0.8% | 30MB | 81.1 |

Table 8: SVD sparsity estimator delivers near-oracle accuracy with negligible computation and memory overheads.

does not employ output token splitting, (2) our proposed approach with output token splitting, and (3) a control configuration, where a random subset of tokens is selected for dense computation, matching the number of output tokens in our approach. The results are presented in Table 9, which demonstrate that selectively preserving dense computation for output tokens consistently outperforms both the random selection and the baseline, highlighting the efficacy of our proposed method in improving computational efficiency without sacrificing performance.

|  | CSR170K | | | Math10K | | |
|---|---|---|---|---|---|---|
| Sparsity | #FL. | Spd. | Acc. | #FL. | Spd. | Acc. |
| All tokens | 100% | 1.0× | 86.7 | 100% | 1.0× | 47.1 |
| Random | 102% | 0.9× | 86.6 | 129% | 0.8× | 70.9 |
| Inputs only | 102% | 0.9× | **86.9** | 129% | 0.8× | **81.1** |

Table 9: Datasets with more output tokens (*e.g.*, Math10K) are highly sensitive to sparsity on those tokens, leading to significant accuracy drops. Output-aware splitting preserves accuracy while still achieving strong runtime improvements.

**Uniform Sparsity without Sensitivity Analysis.** We investigate the impact of applying sensitivity-guided layerwise sparsity compared to uniform sparsity across all model layers. Our experiments, conducted at various speedup targets, show that the sensitivity-aware approach—where sparsity ratios are adapted to each layer's sensitivity—consistently outperforms the uniform sparsity baseline, as demonstrated in Table 10. This result underscores the importance of tailoring sparsification strategies to the unique sensitivity characteristics of each layer, rather than adopting a one-size-fits-all approach. Additionally, Figure 7 illustrates the varying sensitivity of sparsity across different layers, further validating the effectiveness of our layer-wise approach.

| Sparsity Method | #FLOPs | Speedup | Accuracy |
|---|---|---|---|
| Uniform | 60% | 1.1× | 80.3 |
| Nonuniform | 60% | 1.4× | **81.1** |
| Uniform | 46% | 1.5× | 80.2 |
| Nonuniform | 46% | 1.6× | **81.1** |
| Uniform | 37% | 1.6× | 79.5 |
| Nonuniform | 37% | 1.8× | **80.5** |

Table 10: On LLaMA3-8B with Math10K, we show that layerwise sensitivity enables better sparsity allocation. At fixed FLOPs budgets, our non-uniform sensitivity-aware approach outperforms uniform sparsity in speedup and accuracy, achieving a lossless performance up to 1.6× speedup.

|  | TopK Selection Per | | Metric | Accuracy (Math10K) |
|---|---|---|---|---|
|  | Channel | Head |  |  |
| QK Criterion | ✓ | ✗ | Attention Norm | **80.7** |
|  | ✗ | ✓ | Attention Norm | 79.6 |
|  | ✓ | ✗ | L2 Norm | 79.8 |
|  | ✓ | ✗ | Random | 79.1 |
| VO Criterion | – | – | L2Norm | **81.4** |
|  | – | – | Random | 79.6 |
| FFN Criterion | – | – | L2Norm | **81.4** |
|  | – | – | Wanda | 81.3 |
|  | – | – | Random | 78.6 |

Table 11: Comparison of pruning criteria for QK, VO, and FFN modules under 90% uniform sparsity with token splitting at a 5% step offset. All other components are computed densely. For each module, the selected metric: attention norm (channel-wise) for QK, and L2 norm for VO and FFN, achieves the highest accuracy, validating our design choices.

**Pruning Criterion.** We explore pruning criteria beyond L2 norm, such as methods based on weights (Sun et al., 2024). While these approaches show some promise, L2 norm remains the most effective method with simplicity. We compare L2 norm pruning with Wanda (Sun et al., 2024) and random pruning, all using the oracle setting for FFNs. Additionally, we conduct an ablation study on attention projections, pruning heads and channels (same or different per head), as shown in Table 11. Our proposed attention norm performs the best.

## 5. Conclusion

We introduced *SparseLoRA* to accelerate fine-tuning through contextual sparsity using a lightweight, training-free *SVD sparsity estimator*. By dynamically selecting sparse weights for loss and gradient computation, SparseLoRA reduces computational cost by up to **2.2×** and delivers a **1.6×** speedup while preserving accuracy across diverse tasks.

**Acknowledgment.** We would like to thank Google-BAIR Commons, Google DeepMind, and POSCO HOLDINGS for their support of this research. We are also grateful to NVIDIA for providing GPU hardware.

## Impact Statement

This paper presents an approach for compute-efficient fine-tuning of LLMs. There are many potential societal consequences of our work, none of which we feel must be specifically highlighted here.

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

# A. Appendix

## A.1. Analysis of Pruning at Attention Head Level in Inference and Fine-tuning

Liu et al. (2023b) found that some attention heads show uniform attention scores across previous tokens during the auto-regressive generation. As illustrated in Figure 9a, at test time the top head is a uniform "token mixing" head, while the middle and bottom heads are "heavy hitter" heads. Since uniform heads don't capture important interactions, keeping only the heavy hitter heads preserves prediction quality. However, this behavior changes during fine-tuning: Figure 9b shows that attention heads may exhibit different patterns depending on the token—what might be a token-mixing head for one token could be critical for another.

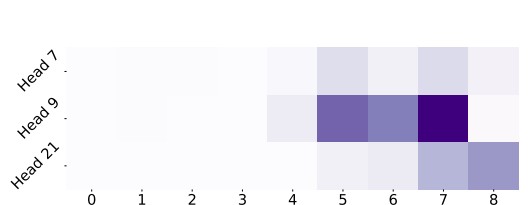
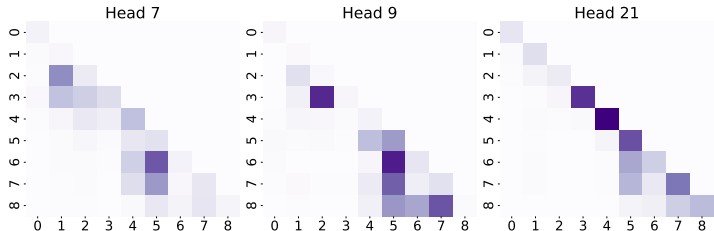

(a) last row in Figure 9b's attention scores visualization    (b) attention scores of 3 different heads for a sample

Figure 9: Attention scores from three different heads are visualized for the last 9 tokens of a sample. Figure 9a (left) corresponds to the last row of Figure 9b (right), simulating the auto-regressive generation of the final token during inference. Darker colors indicate higher attention scores.

## A.2. Fine-Tuning Details

Table 12: Training Hyperparameters Across Datasets. All experiments use LoRA with dropout = 0, rank = 32, and $\alpha = 64$.

| Dataset | Seq. Len | Batch Size | Epochs | LR | Scheduler | Warmup Ratio |
|---|---|---|---|---|---|---|
| CSR170K | 512 | 8 | 1 | 3e-4 | cosine | 0.04 |
| Math10K | 512 | 8 | 3 | 3e-4 | cosine | 0.04 |
| GLUE (COLA, STS-B, RTE, SST2, QNLI, MNLI, QQP) | 128 | 8 | 3 | 5e-5 | cosine | 0.04 |
| GLUE (MRPC, WNLI) | 128 | 8 | 5 | 5e-5 | cosine | 0.04 |
| CodeFeedback | 1024 | 6 | 1 | 2e-5 | cosine | 0.04 |
| WizardLM | 2048 | 2 | 1 | 2e-5 | cosine | 0.04 |

## A.3. Detailed Sparsity Configuration

Table 13: Sparsity Configuration Across Models and Datasets

| Model | Dataset | FFN | | QKVO | | Step | #FLOPs | Speedup |
|---|---|---|---|---|---|---|---|---|
| | | Layers | Sparsity (%) | Layers | Sparsity (%) | | | |
| **LLaMA2-7B** | CSR170K | L13–L29 | 90 | L17–L29 / L20, L24 | 50 | 5% | 0.65 | 1.3× |
| | Math10k | L13–L29 | 90 | L13–L29 / L20, L24 | 60 | | 0.73 | 1.2× |
| | CodeFeedback | L3–L30 | 99 | L14–L19,L21–L23,L25-L29 | 25 | 5% | 0.73 | 1.2× |
| **LLaMA2-13B** | CSR170K | L13–L36 | 97 | L17–L36 | 20 | 10% | 0.61 | 1.3× |
| | Math10k | | | | 25 | 5% | 0.70 | |
| **LLaMA3-8B** | CSR170K | L17–L30 | 97 | L17–L19,L21–L23,L25-L29 | 20 | 5% | 0.65 | 1.3× |
| | Math10k | L3–L30 | 99 | L14–L19,L21–L23,L25-L29 | 75 | | 0.46 | 1.6× |
| | GLUE | L17–L30/L31 | 95/50 | L17–L29 | 75 | 5% | 0.61 | 1.3× |
| **LLaMA3.1-8B** | CodeFeedback | L3–L30 | 99 | L14–L19,L21–L23,L25-L29 | 40 | 5% | 0.66 | 1.3× |
| | WizardLM | | | | | | 0.53 | 1.5× |

## A.4. GaLore vs. SparseLoRA

We compare GaLore (Zhao et al., 2024) and SparseLoRA on CSR170K and Math10K. GaLore training takes slightly more VRAM than LoRA and requires A100 GPUs due to VRAM limitations of A6000 under Distributed Data Parallel (DDP). Runtime is normalized to LoRA, as in our main submission. Result are shown in Table 14. GaLore achieves memory efficiency by projecting the full gradient matrices into a low-rank subspace. This projection is periodically updated during training via an online Singular Value Decomposition (SVD) of the gradients. While this online SVD allows GaLore to adapt the subspace and maintain performance similar to LoRA, it incurs a significant $1.58\times$ training overhead compared to LoRA. The amortized time for GaLore, which accounts for these periodic projection updates via online SVD, substantially slows down the fine-tuning process by $13.72\times$. Thus, while GaLore prioritizes memory-efficient fine-tuning by reducing optimizer states and gradient memory, this comes at a considerable cost to computational efficiency due to the demanding SVD operations. In contrast, SparseLoRA is designed to accelerate fine-tuning while delivering near-lossless performance.

| CSR170K | | | | | | | | | |
|---|---|---|---|---|---|---|---|---|---|
| Model | Runtime | Average | BoolQ | PIQA | SIQA | HellaSwag | WinoG | ARC-E | ARC-C | OBQA |
| LoRA | 1.00 | 87.1 | 74.5 | 89.6 | 82.8 | 95.3 | 88.4 | 93.1 | 84.4 | 88.8 |
| GaLore | 1.58 [13.72] | 84.1 | 71.2 | 87.1 | 79.6 | 92.0 | 85.0 | 89.4 | 80.5 | 87.8 |
| SparseLoRA | 0.78 | 87.0 | 74.7 | 89.5 | 82.8 | 95.3 | 88.8 | 92.9 | 83.6 | 88.3 |

| Math10K | | | | | |
|---|---|---|---|---|---|
| Model | Runtime | Average | GSM8K | SVAMP | MAWPS |
| LoRA | 1.00 | 80.0 | 71.1 | 79.5 | 89.5 |
| GaLore | 1.58 [13.72] | 78.7 | 68.1 | 77.9 | 90.2 |
| SparseLoRA | 0.82 | 80.0 | 70.9 | 79.4 | 89.9 |

Table 14: LLaMA 3-8B on CSR170K and Math10K. GaLore's amortised cost (including periodic online-SVD updates) is shown in brackets.

## A.5. Impacts of Learning Rates Sweep

We perform learning rate sweeps to eliminate selection bias from hyperparameter choices. Specifically, we evaluate both LoRA and SparseLoRA variants using LLaMA3-8B on the Math10K and CSR170K datasets. Table 15 shows that the performance gap between the best-performing LoRA and SparseLoRA is just 0.2% on Math10K and 0.3% on CSR170K, validating the robustness of our approach.

| Dataset | Learning Rate | LoRA | SparseLoRA |
|---|---|---|---|
| Math10K | $3.0 \times 10^{-5}$ | 78.3 | 78.8 |
| | $5.0 \times 10^{-5}$ | 78.6 | 79.3 |
| | $9.5 \times 10^{-5}$ | 79.6 | 79.8 |
| | $3.0 \times 10^{-4}$ | 80.0 | **80.0** |
| | $5.0 \times 10^{-4}$ | **80.2** | 79.6 |
| | $9.5 \times 10^{-4}$ | 78.1 | 77.3 |
| CSR170K | $3.0 \times 10^{-5}$ | 85.7 | 85.6 |
| | $5.0 \times 10^{-5}$ | 86.7 | 86.5 |
| | $9.5 \times 10^{-5}$ | **87.7** | **87.4** |
| | $3.0 \times 10^{-4}$ | 87.1 | 87.1 |

Table 15: Learning-rate sweep of LLaMA 3-8B on Math10K and CSR170K. Best accuracy per dataset is in bold.

## A.6. LoRA on Different Projections

We primarily apply sparsity to accelerate the main branch of the model, while keeping the LoRA branches dense. To assess generality, we also conduct additional experiments on Math10K, applying LoRA to the Q, K, V, up, and down

projections—following the DoRA setup (Liu et al., 2024c)—beyond the configurations explored in the main paper. Results in Table 16 indicate that the benefits of SparseLoRA extend beyond just QKVO, demonstrating its broader applicability.

| Projection Set | LoRA | SparseLoRA | Δ |
|---|---|---|---|
| QKVO | 79.9 | **80.0** | +0.1 |
| QKVUD | 80.3 | **80.9** | +0.6 |
| QKVOGUD | 80.5 | **80.7** | +0.2 |

Table 16: Mean accuracy of LoRA versus SparseLoRA on LLaMA 3-8B for different projection configurations.

### A.7. Iso-FLOP Comparison

A practical question is how methods behave when constrained by a *fixed FLOP budget*, rather than a fixed number of training steps. In production, practitioners often allocate a set amount of compute; a method that extracts more accuracy per FLOP is therefore more valuable. To address this, we perform an Iso-FLOP study on LLaMA3-8B using Math10K and CSR170K. One full epoch is treated as 100% of the available FLOPs, and we sweep down to 5%. Figure 10 shows consistent gains across all tested budgets on both datasets. These results confirm that, for the same computational cost, SparseLoRA produces better-performing models than standard LoRA.

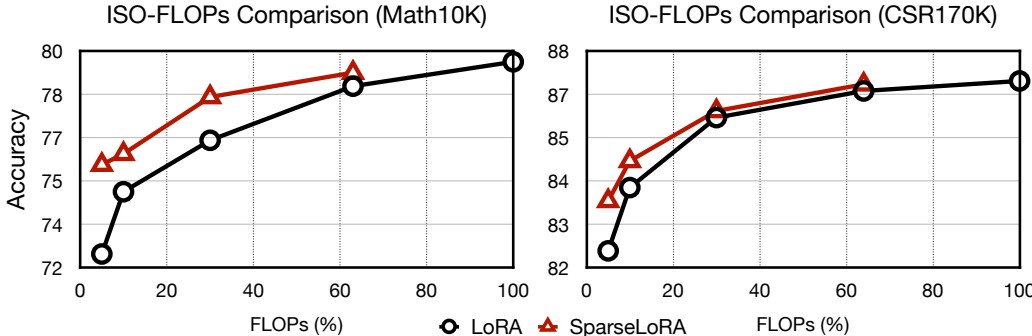

Figure 10: Iso-FLOPs comparison on LLaMA3-8B using Math10K and CSR170K. As the FLOPs budget decreases, SparseLoRA is able to better retain task performance compared to the LoRA counterpart by a larger margin.

