# OpenReview forum: "SparseLoRA: Accelerating LLM Fine-Tuning with Contextual Sparsity"
_ICML.cc/2025/Conference — ICML 2025 poster_

### Official Review · Reviewer_j9K3 · 2025-03-13

**Overall Recommendation:** 3

**Summary:**

The paper proposes to accelerate LoRA fine-tuning with contextual sparsity. Tailored for fine-tuning, they propose a lightweight, training-free SVD sparsity estimator to reduce computation overhead. Experimental results show that they can speed up LoRA fine-tuning by 1.4x.

**Claims And Evidence:**

Yes

**Essential References Not Discussed:**

"S2FT: Efficient, Scalable and Generalizable LLM Fine-tuning by Structured Sparsity" is a recent paper published in Arxiv in December. However, it is officially published in ICLR 25 after the submission deadline of ICML.

**Experimental Designs Or Analyses:**

Yes

**Methods And Evaluation Criteria:**

Yes

**Other Comments Or Suggestions:**

April 12: I will keep my score.

**Other Strengths And Weaknesses:**

Strengths
- I think it is an interesting and meaningful observation that output tokens are more sensitive to pruning.

Weaknesses
- I can not see why this method needs to be used together with LoRA. It would be great to use it independently and compare it with FFT.

**Questions For Authors:**

No

**Relation To Broader Scientific Literature:**

The motivation is related to contextual sparsity but the key contributions are how to apply the idea to fine-tuning.

**Theoretical Claims:**

Not applicable.

---

> ### Author Rebuttal · Authors · 2025-04-01
>
> We sincerely appreciate the reviewer’s thoughtful feedback and recognition of our approach. Below, we address each of the concerns in detail:
>
> > "S2FT: Efficient, Scalable and Generalizable LLM Fine-tuning by Structured Sparsity" is a recent paper published in Arxiv in December. However, it is officially published in ICLR 25 after the submission deadline of ICML.
>
> We would like to clarify that our paper already cites the S2FT work. S2FT introduces a structured pruning approach aimed at memory-efficient fine-tuning. According to Figure 5 in their paper, S2FT achieves a speedup of 1.1× over LoRA on LLaMA2-7B for the Commonsense 170K dataset. In contrast, our method achieves a 1.32× speedup on the same setting. While S2FT prioritizes memory efficiency and accuracy, SparseLoRA addresses a complementary direction: computational efficiency with lossless accuracy. We will include a more detailed discussion of S2FT in the final version of the paper.
>
>
> > I can not see why this method needs to be used together with LoRA. It would be great to use it independently and compare it with FFT.
>
> We agree that extending our method beyond LoRA fine-tuning is an exciting direction. However, there are some practical considerations tied to the characteristics of LoRA. A key component of our method is the dynamic prediction of activation sparsity using an SVD-based predictor. This predictor relies on the static nature of the base layer weights—which, in LoRA, remain frozen during fine-tuning. In contrast, in full fine-tuning settings, the base weights are updated throughout training. This undermines the validity of the SVD-based predictor, potentially leading to inaccurate sparsity estimates. At this stage, our method leverages the frozen base layers in LoRA to reliably identify and exploit structured sparsity for speedup. Exploring how to adapt the predictor dynamically for full fine-tuning is an interesting direction we plan to pursue in future work.

---

### Official Review · Reviewer_pTrd · 2025-03-15

**Overall Recommendation:** 3

**Summary:**

The paper introduces a method to accelerate fine-tuning of large language models (LLMs) by leveraging contextual sparsity. Unlike existing parameter-efficient fine-tuning (PEFT) methods such as LoRA and DoRA, which reduce memory usage but not computational cost, SparseLoRA optimizes both memory and computation.
The key contributions of the paper include: a training-free SVD-based sparsity estimator that selects a subset of weights for loss and gradient computation, context-aware sparsity application (non-uniform sparsity across layers based on sensitivity analysis, selective sparsity for context tokens, while keeping output tokens dense, progressive sparsity).

Empirical validation on commonsense and arithmetic reasoning benchmarks is provided.

**Claims And Evidence:**

* While the computation speedup is clear, the memory analysis and comparison to the baselines (presented by the authors and not presented, please see below) are missing.
* The comparison of the proposed method was done to a narrow set of models: LoRA and DoRA while there are more baselines that could be compared (please see my points below).

The claims regarding training speed-up are supported by empirical results.

**Essential References Not Discussed:**

The paper cites enough related papers.

**Experimental Designs Or Analyses:**

There are plenty of works in LLM PEFT domain. I think the most missing baselines are APT[1] and Galore[2] methods (or their variants). Also I found [3] that looks similar to the proposed method. The authors presented only  2 baselines

Despite the lack of additional baselines, I think that experimental design in terms of datasets and metrics is reasonable.


[1] Zhao, Bowen, Hannaneh Hajishirzi, and Qingqing Cao. "APT: adaptive pruning and tuning pretrained language models for efficient training and inference." Proceedings of the 41st International Conference on Machine Learning. 2024.

[2] Zhao, Jiawei, et al. "GaLore: Memory-Efficient LLM Training by Gradient Low-Rank Projection." International Conference on Machine Learning. PMLR, 2024.

[3] Huang, Weizhong, et al. "Dynamic Low-Rank Sparse Adaptation for Large Language Models." The Thirteenth International Conference on Learning Representations.

**Methods And Evaluation Criteria:**

The benchmark datasets choice looks reasonable to me. However I would like to see the results for a commonly used GLUE benchmark and not only common and arithmetic reasoning.

**Other Comments Or Suggestions:**

* I think the format of the paper is not the same as provided in the ICML 2025 template (e.g. tables captions should be above the tables), please fix it.

**Other Strengths And Weaknesses:**

**Strengths**:
* The proposed model reduces computation time by sparsifting the base model weights during the finetuning in addition to the low-rank adaptor training.
* the method is evaluated on commonsense and arithmetic reasoning
* The method reduces the number of FLOPS by 30-40%
* runtime breakdown of LLM fine-tuning and sensitivity analysis on layer-wise sparsity is interesting and important.

**Weaknesses**:

* The proposed method  doesn’t improve the finetuned model accuracy but only reduces the training time

* The GPU memory increase/decrease is not discussed.
* Missing a comparison to the optimizer-base finetuning methods such as a Galore or other recent methods in terms of training time and GPU memory.
* Missing more baselines for sparse finetuningof LLMs

**Questions For Authors:**

* I don't find the values of k in svd decomposition you used in your experiments / or sparsity levels in Tables 1 + 2
* It is not clear from the text what is your **inference** setup and how the inference times are compared across baselines and your method? I see only the training time measurements.

* Why are some experimental results missing for DoRA method? (E.g. with base model LLaMA2-13B)

**Relation To Broader Scientific Literature:**

The method builds upon and extends:
* LoRA (Hu et al., 2022) and DoRA (Liu et al., 2024b) for parameter-efficient fine-tuning.
* Contextual sparsity approaches in LLM inference (Liu et al., 2023).

The key novelty is applying structured contextual sparsity to fine-tuning, whereas previous methods focused on inference-time acceleration. The work is highly relevant to ongoing research in:
 * Efficient LLM training (Thangarasa et al., 2023, Mozaffari et al., 2024)
 * Sparse computing for neural networks (Han et al., 2015, 2016)

The paper appropriately cites relevant work but does not discuss:
* Alternative structured sparsity methods (e.g., block sparsity, hardware-aware sparsity).
* Recent advances in mixed low-rank and sparse fine-tuning methods (e.g., WeLore, SLoPe).

**Theoretical Claims:**

There are no theoretical claims

---

> ### Author Rebuttal · Authors · 2025-04-01
>
> > While the computation speedup is clear, the memory analysis and comparison to the baselines are missing
>
> Our approach uses LoRA for fine-tuning, so the memory profile remains the same as LoRA. Sparsifying the main branch does not affect memory usage.
>
> > Include comparisons with additional PEFT methods like APT and LoSA
>
> APT and LoSA are designed to optimize inference-time sparsity rather than fine-tuning speed. Their methods typically slow down fine-tuning significantly to produce a sparse model with minimal accuracy loss at inference. For instance, LoSA requires 45.34 minutes compared to 13.78 minutes for LoRA, and under the Wanda setting, it takes 73.91 minutes versus 21.40 minutes for LoRA under SparseGPT, as shown in Table 8 of the original paper.
>
> > Include comparisons with additional PEFT methods like GaLore. Missing comparison to GaLore in terms of training time and GPU memory
>
> We compare LoRA, GaLore, and SparseLoRA on Commonsense 170K and Math 10K. GaLore training requires A100 GPUs due to VRAM limitations of A6000 under DDP. Runtime is normalized to LoRA, as in our main submission. GaLore incurs a 1.58x training overhead but achieves similar performance to LoRA. The amortized time of GaLore, accounting for projection updates and online SVD, slows fine-tuning by 13.72x. While GaLore focuses on memory-efficient fine-tuning at the cost of computational efficiency, SparseLoRA accelerates fine-tuning with near-lossless performance.
>
>
> ### LLaMA3-8B Commonsense 170K [on A100s]
>
> |Model|Runtime|Mean|BoolQ|PIQA|Social-IQA|HellaSwag|Winogrande|ARC-Easy|ARC-Challenge|OpenBookQA|
> |-----|-------|----|-----|-----|----------|---------|----------|--------|--------------|----------|
> |LoRA|1.00|87.1|74.5|89.6|82.8|95.3|88.4|93.1|84.4|88.8|
> |GaLore|1.58[13.72]|84.1|71.2|87.1|79.6|92.0|85.0|89.4|80.5|87.8|
> |SparseLoRA|0.78|87.0|74.7|89.5|82.8|95.3|88.8|92.9|83.6|88.3|
>
> ### LLaMA3-8B Math 10K [on A100s]
>
> |Model|Runtime|Mean|gsm8k|svamp|mawps|
> |-----|-------|----|-----|-----|----------|
> |LoRA|1.00|80.0|71.1|79.5|89.5|
> |GaLore|1.58[13.72]|78.7|68.1|77.9|90.2|
> |SparseLoRA|0.82|80.0|70.9|79.4|89.9|
>
> > Include results on GLUE benchmark
>
> Following the reviewer's recommendation, we extend LoRA and SparseLORA using Llama3-8B on the GLUE benchmark. SparseLoRA maintains competetive perofrmance on sequence classification accross various subsets of the GLUE benchmark.
>
> ### LLaMA3-8B GLUE Benchmark
>
> |Model|Mean|MRPC|SST2|QNLI|RTE|QQP|COLA|
> |-----|----|----|----|----|---|---|-----|
> |LoRA|88.6|92.1|96.2|95.2|88.8|91.8|67.7|
> |SparseLoRA|88.6|92.3|96.4|95.5|88.7|91.9|66.7|
>
> > The proposed method doesn’t improve the finetuned model accuracy but only reduces the training time
>
> Our method is designed to accelerate fine-tuning while preserving accuracy—not to improve accuracy over standard fine-tuning methods.
>
>
> > The GPU memory increase/decrease is not discussed
>
> SparseLoRA improves computational efficiency during fine-tuning without altering the memory usage compared to baseline LoRA.
>
>
> > Specify SVD rank (k) values and sparsity levels used in experiments
>
> We use an SVD Rank of 8 across all models and datasets, resulting in minimal runtime overhead (Table 3). Layer sparsity ratios are based on sensitivity analysis per model, with Llama2-7B's layer-wise analysis shown in Figure 7. The specific sparsity ratios employed are:
>
> ### Model Sparsity Information
>
> |Model|Dataset|FFNSparsity|FFNSparseLayers|QKVOSparsity|QKVOSparseLayers|
> |-----|-------|-----------|---------------|-------------|----------------|
> |Llama2-7B|Commonsense170K|90|L13-L29|60|L17-L29/L20,L24|
> |Llama2-7B|Math10k|90|L13-L29|60|L13-L29/L20,L24|
> |Llama2-13B|Commonsense170K|90|L17-L37|60|L17-L37|
> |Llama2-13B|Math10k|90|L13-L37|60|L13-L37|
> |Llama3-8B|Commonsense170K|90|L13-L29|60|L15-L29|
> |Llama3-8B|Math10k|90|L9-L29|60|L9-L29|
>
> The selected layers and sparsity rates remain mostly consistent across models and datasets. Sensitivity analysis identifies layers unsuitable for sparsity (i.e. layers 20 and 24 in Llama2-7B). On Math 10k, layer ranges were increased to account for token splitting overhead. The sparsity assignment is latency-driven and informed by sensitivity analysis, ensuring no added overhead in SparseLoRA.
>
> > Inference Setup
>
> SparseLoRA targets fine-tuning accleration and only applies sparsity during fine-tuning; inference remains unchanged compared to baseline LoRA.
>
> > Missing DoRA Results
>
> Results for the DoRA method with LLaMA2-13B are missing due to OOM on A6000 GPUs without gradient checkpointing, as DoRA requires significantly more memory.
>
> > Template Issues
>
> We will correct table captions and formatting in the revised version.

---

> > ### Comment · Reviewer_pTrd · 2025-04-04
> >
> > After reviewing the authors' rebuttal and the additional results, I would like to revise my evaluation of the paper positively. Furthermore, I believe that releasing the code is essential for the research community.

---

> > > ### Author Response · Authors · 2025-04-05
> > >
> > > Thank you for the positive feedback and updated evaluation! We appreciate your thoughtful suggestions—they were instrumental in improving the clarity and completeness of our work. We will incorporate the additional results and feedback into the revised manuscript. We also fully agree that releasing the code is essential, and upon acceptance, we are committed to open-sourcing clean, well-documented code to support adoption and further research on SparseLoRA.

---

### Official Review · Reviewer_NtQC · 2025-03-20

**Overall Recommendation:** 3

**Summary:**

Previous parameter-efficient fine-tuning (PEFT) methods, such as LoRA and its variants, have primarily focused on memory efficiency and lightweight storage. However, these approaches do not necessarily lead to faster fine-tuning. This paper introduces SparseLoRA, a novel technique that accelerates fine-tuning by selecting a sparse subset of the base model’s weights, enabling more efficient loss and gradient computation while fully preserving LoRA’s structure.

SparseLoRA achieves this by decomposing the original weight matrix and selectively activating channels based on a Singular Value Decomposition (SVD) sparsity estimator. This estimator adaptively determines sparsity using certain norms of the batched input, allowing the method to remain dynamic and data-aware. Unlike prior works that apply sparsity only during inference (typically with a batch size of 1), SparseLoRA incorporates sparsity directly into the training process.

During training, LoRA initially operates in its standard dense form for the first few iterations. Sparse fine-tuning is then gradually introduced, reducing the computational overhead while maintaining performance. Experimental results demonstrate that SparseLoRA achieves faster training times with minimal accuracy degradation across multiple benchmark tasks, making it a promising alternative for efficient fine-tuning.

**Claims And Evidence:**

While the claims in the paper are generally well-supported, some unconvincing weaknesses remain. Below, I outline key concerns that, if addressed, would strengthen the paper’s experimental robustness:

1. The experiments use a fixed learning rate, but it is standard practice to perform a hyperparameter sweep and report the best-performing configuration. To ensure fair comparisons, Tables 1 and 2 should reflect results from a learning rate sweep, rather than relying on a single fixed value. Otherwise, there is a risk that SparseLoRA benefits simply from better tuning rather than intrinsic efficiency.

2. The paper evaluates SparseLoRA only on QKVO projections. However, LoRA can be applied to different subsets of projections, and it is unclear whether SparseLoRA remains effective across different configurations. A more thorough evaluation should test SparseLoRA on different subsets of trainable LoRA projections to confirm that its benefits generalize beyond QKVO.

3. A crucial missing experiment is an Iso-FLOP comparison—i.e., comparing models trained with the same computational budget. In real-world applications, practitioners often have a fixed FLOP budget rather than a fixed number of iterations. Therefore, it is important to test whether LoRA trained with the same FLOP budget as SparseLoRA produces weaker models.

In Table 1 (LLaMA3-8B results), SparseLoRA trains at 0.62x FLOPs of full LoRA. The paper should compare whether training LoRA for the same 0.62x FLOPs produces weaker performance than SparseLoRA. The paper should include a graph of benchmark accuracy vs. wall-clock time (or training FLOPs). This would help practitioners determine when SparseLoRA is beneficial and when standard LoRA suffices.

While SparseLoRA presents a promising approach for efficient fine-tuning, the paper lacks critical experimental validations. A more rigorous study should incorporate learning rate sweeps to avoid selection bias, evaluations across different LoRA projection sets, and, most importantly, Iso-FLOP comparisons and efficiency trade-offs, as practitioners need to know whether SparseLoRA is truly advantageous under fixed compute constraints. Without these, the claims of SparseLoRA’s efficiency remain incomplete and may not fully guide real-world adoption.

**Essential References Not Discussed:**

Not that I know of.

**Experimental Designs Or Analyses:**

The experiments seem sound. However, given the empirical nature of the paper, necessary experimental details are missing, especially those one would usually find in the supplementary material. For instance, I cannot find more information on Table 1 when I want to check how many shots were used for each task, or the absolute runtime in terms of wall-clock time, or whether FLOPs includes SVD or not.

**Methods And Evaluation Criteria:**

Yes.

**Other Comments Or Suggestions:**

- It would be more helpful if Figure 1 is more descriptive, e.g., which models, tasks, and datasets were used for the particular experiment.
- In lines 43-44, when claiming "adding less than 0.5% overhead to finetuning", does this mean computationally (if so, in terms of FLOPs or wall-clock time) or memory (VRAM)?
- In general, the paper is lacking in detail, e.g., experimental settings, captions in figures. The paper will be stronger with these details.

**Other Strengths And Weaknesses:**

**Strength:**
- Computationally efficient finetuning methods are not well developed. This work attempts to achieve this without sacrificing accuracy. The motivation is extremely practical and the sparse selection mechanism is sensible. However, I am not particularly well-versed in this sub-area, so I am not entirely sure how it fares to prior work or if this is truly the first work to consider computationally efficient PEFT methods.
- Adaptive sparsity that takes into account training-time input batches probably requires more care than dealing with just one input as in inference-time.

Weaknesses listed "Claims And Evidence" and "Questions For Authors".

**Questions For Authors:**

Listed in order of decreasing importance.
1. **Details on SVD.**
I am curious to know the details on the computation of SVD that is done offline, such as time, memory, and etc. Furthermore, when finetuning with SparseLoRA, how much memory overhead is included by loading the SVD decomposition alongside the model weights? Include details about GPU memory in Table 1 seems important even if the overhead is minimal.
2. **Motivation for computationally efficient PEFT.**
Thought PEFT methods are not computationally efficient, often this is not much of a concern since tasks that require finetuning do not require massive datasets and thus does not consume long GPU hours. While potentially useful, I have some reservations whether such method is necessary. Can the authors provide examples in which PEFT would take more than hundreds of GPU hours, making computationally efficient PEFT methods necessary? In my practical experience, I'd rather train hours to a day more if that means I do not sacrifice accuracy for finetuning tasks.
3. **DoRA Settings.**
DoRA was first published as an improvement to LoRA, albeit the additional overhead in computation. I assume there must be settings in which DoRA is better than LoRA, as settings described in the DoRA paper. Could authors comment on whether such setting exists, and if SparseLoRA's performance still holds in that particular setting as well?

With the weaknesses and questions addressed appropriately, I am willling to change my evaluation of the paper. Thank you.

**Relation To Broader Scientific Literature:**

- Computationally efficient finetuning methods are not well developed. This work attempts to achieve this without sacrificing accuracy. The motivation is extremely practical and the sparse selection mechanism is sensible. However, I am not particularly well-versed in this sub-area, so I am not entirely sure how it fares to prior work or if this is truly the first work to consider computationally efficient PEFT methods.
- The SVD estimator based on two kinds of norm-based criteria can be used elsewhere that require input-adaptive sparsity.

**Theoretical Claims:**

N/A.

---

> ### Author Rebuttal · Authors · 2025-04-01
>
> We appreciate the reviewer’s comments:
>
> > Fixed learning rate instead of hyperparameter sweeps.
>
> For concision we only include mean performance. Here V1 refers to results in Table 1-2 of the paper and V2 is the new "conservative" config in our response to Reviewer as16
> ### LLaMA3-8B LR Sweep (Math10K)
> |LearningRate|LoRA|SparseLoRA-V1|SparseLoRA-V2|
> |-|-|-|-|
> |3.00E-05|78.3|77.7|78.8|
> |5.00E-05|78.6|78.6|79.3|
> |9.49E-05|79.6|79.2|79.8|
> |3.00E-04|80.0|**79.8**|**80.0**|
> |5.00E-04|**80.2**|79.3|79.6|
> |9.49E-04|78.1|77.1|77.3|
>
> ### LLaMA3-8B LR Sweep (Commonsense 170K)
> |LearningRate|LoRA|SparseLoRA-V1|SparseLoRA-V2|
> |-|-|-|-|
> |3.00E-05|85.7|84.8|85.6|
> |5.00E-05|86.7|85.9|86.5|
> |9.49E-05|**87.7**|**86.8**|**87.4**|
> |3.00E-04|87.1|86.3|87.1|
>
> We conduct two learning rate sweeps with Llama3-8B on Math10K and CSR170K for LoRA and SparseLoRA. SparseLoRA shows minimal performance degradation compared to LoRA, confirming our preformance claims. In the conservative setting, the performance gap between the optimal LoRA and SparseLoRA is 0.2% on Math10K and 0.3% on CSR170K.
> > SparseLoRA is only evaluated on QKVO projections, but LoRA can be applied to different subsets of projections.
>
> We focus on applying sparsity to speed up the main branch of the model, leaving the LoRA branches unchanged. However, we also provide additional experiments applying LoRA to Q, K, V, up, and down projections, following DoRA, in addition to the settings in our paper.
> ### LLaMA3-8B Performance on Math10K for different projections
> |Method|Config|Mean|
> |-|-|-|
> |LoRA|QKVO|79.9|
> |SparseLoRA|QKVO|80.0|
> |LoRA|QKVUD|80.3|
> |SparseLoRA|QKVUD|80.9|
> |LoRA|QKVOGUD|80.5|
> |SparseLoRA|QKVOGUD|80.7|
> > Provide iso-FLOP comparisons showing performance vs. computational budget
>
> Following the reviewer’s recommendation, we conduct an iso-flop comparison on Llama3-8B using Math10K and CSR170K. SparseLoRA outperforms LoRA, with a 3% improvement at 5% FLOP budget on Math10K. Experiments run using 1 epoch as 100% FLOPs.
> ### Iso-FLOP Comparison on Math10K with LLaMA3-8B
> |Setting|FLOP(%)|Mean|Diff|
> |-|-|-|-|
> |**LoRA (Baseline)**|100|79.6|--|
> |**LoRA**|63|78.7|--|
> |**SparseLoRA**|63|79.2|+0.55|
> |**LoRA**|30|76.7|--|
> |**SparseLoRA**|30|78.3|+1.55|
> |**LoRA**|10|74.8|--|
> |**SparseLoRA**|10|76.2|+1.63|
> |**LoRA**|5|72.5|--|
> |**SparseLoRA**|5|75.8|+3.30|
> ### Iso-FLOP Comparison on Commonsense 170K with LLaMA3-8B
> |Setting|FLOP(%)|Mean|Diff|
> |-|-|-|-|
> |**LoRA (Baseline)**|100|87.1|--|
> |**LoRA**|64|86.8|--|
> |**SparseLoRA**|64|87.0|+0.28|
> |**LoRA**|30|86.0|--|
> |**SparseLoRA**|30|86.2|+0.19|
> |**LoRA**|10|83.9|--|
> |**SparseLoRA**|10|84.7|+0.83|
> |**LoRA**|5|82.0|--|
> |**SparseLoRA**|5|83.5|+1.47|
>
> > Lack of specifics like number of shots per task etc.
>
> Tasks are evaluated with the mean over 5 shots. The SVD sparsity estimator is used during training to skip parts of the base-branch computation. The SVD computation is offline, but during training, marginal FLOPs are spent on the sparsity metric, accounting for 0.0534% of LoRA FLOPs and 0.8% runtime overhead, as shown in Table 3.
>
> > Details on SVD Predictor
>
> We compute the SVD of weight matrices to create our sparsity estimator (Algorithm 1, lines 251-254) once per model. For the self-attention block, we SVD q, k, and v, and for the FFN, we compute up and gate projections, as described in Section 3.1. SVD is performed with rank 8 for all models. The predictors incur minimal runtime during training (Table 3) and negligible memory overhead (see below in Mb).
> - Llama2-7B: 26.8
> - Llama2-13B: 36.9
> - Llama3-8B: 30.0
>
> > Motivation for computationally efficient PEFT
>
> SparseLoRA achieves near-lossless performance while reducing computation. Even if PEFT tasks are fast, practitioners favor any free acceleration. For example, many opt for BF16 training over FP32 because, despite its slightly lower precision, the efficiency gains are substantial and the performance drop is negligible. Faster training means more rapid experimentation and ability to scale to larger models or more complex tasks—all of which are significant benefits in practice. SparseLoRA is a step toward this goal and can inspire future improvements in computationally efficient, lossless PEFT methods.
>
> > DoRA Settings
>
> In this work, we use a rank of 32 for LoRA fine-tuning; DoRA has shown a smaller performance gap with this setting (Figure 5 in their work). S2FT reproduced similar results on Llama3-8B (arithmetic).  Unlike DoRA—which prioritizes stability and performance at the expense of runtime—SparseLoRA enhances efficiency, complementing DoRA for practitioners.
>
> > Figure 1
>
> We visualize the results directly from Table 1, specifically highlighting the normalized runtime and average performance on CSR170K for Llama2-7B.
>
> > "less than 0.5% overhead to finetuning" by SVD
>
> We meant "0.8%" runtime overhead introduced by SVD estimator. Additional metrics in Table 3.

---

> > ### Comment · Reviewer_NtQC · 2025-04-04
> >
> > Thank you for the detailed rebuttal. I now have more confidence in the experiments, especially due to the Iso-FLOPs experiments. I hope the authors release clean reusable code (if accepted) for wider adoption and examination of their method. I have adjusted my score accordingly (2->3).

---

> > > ### Author Response · Authors · 2025-04-05
> > >
> > > Thank you for the positive feedback and score adjustment! We’re glad the experiments clarified our contributions. We’ll incorporate these results and other suggestions into the revised manuscript. Upon acceptance, we’re committed to releasing clean, well-documented code to support broad adoption and further exploration of SparseLoRA.

---

### Official Review · Reviewer_as16 · 2025-03-21

**Overall Recommendation:** 3

**Summary:**

The paper proposes a framework for accelerating the fine-tuning large language models by structured pruning of pretrained weight matrices, and using dense and trainable LoRA adapters. The core proposed idea is to estimate the importance of each channel in a pretrained weight matrix, prune the unimportant channels by slicing the weights of that channel, and then using the sliced weights for computation. The LoRA computation is done following the original LoRA paper, with dense matrices.

The porposed method uses different pruning strategies for different components:
- FFN: Uses the L2 norm of the activations of the Gate projection after SiLU activation along the batch and sequence dimensions, and retains the columns in the Gate and Up weight matrices with the highest L2 norm (assuming columns represent the weights if a single channel). The corresponsing rows of the Down weight matrix are also pruned.
- Value and Output Projection in the attention layer: Uses the L2 norm of the activations of the attention heads after the Value projection along the batch and sequence dimensions, and retains the columns in the Value matrix with the highest L2 norm. The corresponding rows of the Output matrix are also pruned.
- Query and Key: Calculates the L2 norm of the Key and Value projections along the batch and sequence dimensions, and uses their dot product as the pruning criterion. The columns in the Key and Query matrices with the highest dot product are retained.

The paper proposes several tricks to reduce computational cost and improve performance:

- T1: The paper also proposes an SVD based estimator for the sparsity. Instead of using the projections from full pretrained weight matrices in order to calculate the L2 norms (the sparsity metric), the operations are performed using the forst $k$ ranks of their respective SVD projections, which results in reduced computational cost for the calculation of the sparsity metric.

- T2: The sparsity ratio is selected based on layer sensitivity analysis, with higher sparsity for deeper layers. To determine the sparsity ratio, the authors perform a sensitivity analysis by greedily increasing the sparsity ratio of each layer while keeping the other layers dense, and measuring the performance on a subset of the Commonsense Reasoning tasks.

- T3: The pruning (sparsity) is only applied to the context tokens, and dense pretrained weight matrices are retained for the output tokens.

- T4: Early iterations in fine-tuning are run in dense mode (upto 20% of the total finetuning iterations) and sparsity is imposed in the later iterations.

- T5: Uses sequence averaging in the SVD estimator for the FFN block to reduce the computational cost. *However, the authors do not define what they mean by the term, and do not provide the exact implementation details. See Point 5 in the Strengths and Weaknesses section.*

## Update after rebuttal

I have gone through the authors' rebuttal and their response to other reviews. I do not mind raining my score to 3, provided the writing of the paper is improved based on my comments in the Other Strengths And Weaknesses section.

**Claims And Evidence:**

1. **Sparsity metric**: The use of L2 norm is well justified, but I would like to mention some nuances to the results of the ablation studies in Table 6 which compares L2 norm with Random pruning and Wanda: In case of the Self-Attention block, L2 norm does not show a significant improvement over Random pruning. However, in the FFN block, L2 norm shows an improvement over Random pruning and maintins simplicity. The authors should mention this in the paper.

2. **Channel pruning vs. Head pruning**: Section 3.1.2, mentions that head level pruning strategy could be problematic. However, Table 6 shows that the accuracy with Head pruning is 80.1 and accuracy due to channel pruning is 80.2, which is not a significant difference. Hence, the choice of channel pruning over head pruning proposed in the paper is not very well justified.

**Essential References Not Discussed:**

N/A

**Experimental Designs Or Analyses:**

The design of the experiments and results reflect the goal if the paper. However, the ablation studies need to be described in more detail. Specifically, when the checking the effect of a particular setting, what are the other settings that are kept constant? For example, Table 5 shows the effect of uniform vs. layer specific sparsity. What are the other settings that are kept constant? The paper should provide a detailed description of the ablation studies.

**Methods And Evaluation Criteria:**

The paper evaluates the proposed SparseLoRA on Commonsense170k and Math10k benchmarks, which have been used in prior works [Hu et al. (2023), Liu et al. (2024)]. The performance on a task is measured using the accuracy of the final answer, which is standard in prior PEFT works. The computaitonal efficiency is measured using the wall clock time and the number of FLOPs relative to LoRA fine-tuning, which reflects the goal of the paper.

---

## References
[1] Zhiqiang Hu, Lei Wang, Yihuai Lan, Wanyu Xu, Ee-Peng Lim, Lidong Bing, Xing Xu, Soujanya Poria, Roy Lee, "LLM-Adapters: An Adapter Family for Parameter-Efficient Fine-Tuning of Large Language Models", EMNLP 23

[2] Shih-Yang Liu, Chien-Yi Wang, Hongxu Yin, Pavlo Molchanov, Yu-Chiang Frank Wang, Kwang-Ting Cheng, Min-Hung Chen, "DoRA: Weight-Decomposed Low-Rank Adaptation", ICML 24

**Other Comments Or Suggestions:**

N/A

**Other Strengths And Weaknesses:**

## Strengths
- Achieves 1.7x reduction in computational cost and 1.4x reduction in wall clock time over LoRA, while achieving almost the same accuracy in most cases.
- The paper proposes a novel method for estimating the sparsity metric using SVD decomposition of the pretrained weights, which reduces the computational cost of the sparsity metric calculation.

## Missing hyperparameter details
1. The expeirmental details for the SVD rank and the exact sparsity ratios per layer are missing.

## Writing and Clarity
The writing is unclear in many places:

2. In section 4.3 (Analysis), none of the experiments and tables mention the model being used, or how the accruracy on Math10k is calculated (for e.g., is it the mean of the accuracies of individual datasets?). Furthermore, the accuracies on Math10k in the Tables 3, 4, 5, and 6 are much higher than the accuracies reported in Table 2. What is the reason for this discrepancy?

3. Figure 8 seems to contradict the text on line 245, which claims that sparsity is only applied to the context tokens, and output tokens are processed in a dense manner. However, the figure shows that only a small portion of the output tokens use dense weights.

4. In Table 5, what do columns 1.4x Speedup and 1.6x speedup denote, and how do they relate to the main results in Table 2? Table 5 also does not mention what benchmarks the experiments are performed with (from the accuracy scores, it looks to be Math10k)

5. The paper is unclear about the definition and implementation of the sequence averaging in the SVD estimator.

6. The paper is very fragmented in terms of what optimizations lead to the reported FLOPs and Runtime decrease. Lines 379, Section 4.3, mentions that sequence averaging incurs leads to significant speedup, yet it has not been defined or explained in the paper. A coherent explanation of exactly which optimizations lead to the decrease in FLOPs and Runtime reported in Tables 1 and 2 should be provided.

**Questions For Authors:**

1. To determine the sparsity ratio for every layer, the authors perform a sensitivity analysis by greedily increasing the sparsity ratio of each layer while keeping the other layers dense. Is the model trained to convergence for every setting?

2. Related to Q1, if the model is trained for every combination, it will add to the computaitonal cost. Section 3.3 studies this with Llama2-7B and a subset of Commonsense Reasoning. Is the same sparsity ratio used across all the models and tasks? If the same ratio is applicable to all models and tasks, then the sensitivity analysis could be done once and the same ratio could be used across all models and tasks. If not, then the sensitivity analysis needs to be done for each model and task, which could be computationally expensive, adding to the overall computational cost of the method.

**Relation To Broader Scientific Literature:**

1. Using sparisity and structured pruning during finetuning has been explored before by Ma et al. (2024), which uses structured pruning along with full fine-tuning of unpruned model parameters. Ma et al. (2024) have also experimented with L2 norm as a sparsity metric, along with other metrics. Their findings show that L2 norm performs slightly worse than other metrics.

2. This paper extends the idea of pruning+finetuning to use adapters (LoRA) instead of full fine-tuning, and uses SVD decomposition of the pretrained weights to obtain the sparsity metrics for efficiency. This paper also proposes channel pruning instead of head pruning in the self-attention block. However, as mentioned in the Claims and Evidence section (Point 2), the contribution of channel pruning over head pruning is not very significant.

3. In terms of the efficiency metrics, while the proposed method can achieve a reduction in FLOPs to ~60-80% that or LoRA, the accuracy is not guaranteed to stay within a small range, suggesting that there could be limitations to the practical applicability of the proposed method. For example, on the Math10k benchmark (Table 2), with LLaMA2-7B, the performance drop on SVAMP is aaround 5% compared to LoRA. With Llama2-13B, the performance drop on GSM8k and SVAMP is around 2%. With Llama3-8B, the drop on GSM8k is ~2%. This drop in performance is not very significant, but it is not negligible either. LoRA, on the other hand, consistently achieves higher accuracy, while being very practical in terms of real-world applications.

Overall, the contribution of the paper is fairly incremental and given the comparison with LoRA mentioned above, the practical applicability of the proposed method could be limited.

---

## References
[1] Da Ma, Lu Chen, Pengyu Wang, Hongshen Xu, Hanqi Li, Liangtai Sun, Su Zhu, Shuai Fan, Kai Yu, "Sparsity-Accelerated Training for Large Language Models", ACL Findings 2024

**Theoretical Claims:**

N/A

---

> ### Author Rebuttal · Authors · 2025-04-01
>
> We sincerely appreciate the reviewer's comments.
>
> > L2 norm vs. Random pruning in Self-Attention blocks
>
> L2 norm shows clear benefits for FFN blocks, while its gains over Random pruning in Self-Attention blocks are modest. We use a unified L2-based criterion to avoid over-engineering and ensure broad applicability.
>
> > Channel vs. Head Pruning:
>
> Although the performance gap is small (80.2 vs. 80.1), channel pruning offers greater flexibility. Unlike head pruning—which enforces equal channel counts—channel pruning enables nuanced selection of important channels, allowing precise control over computational cost.
>
> >SVD Rank and Per-Layer Sparsity Ratios:
>
> For all models, we use an SVD rank of 8, which incurs marginal overhead (see Table 3). Our per-layer sparsity ratios are determined via sensitivity analysis per model (see Figure 7). For the detailed configurations, please check "Model Sparsity Information table" in our response to Reviewer pTrd. Similar settings, with slight adjustments for latency, are applied across models.
>
> > Model Specification and Math10K Accuracies
>
> All experiments in Section 4.3 use LLaMA3-8B. The Math10K results for LLaMA3-8B in Table 2 have 79.8% accuracy on average. This matches the result in line 2 of Table 3, line 2 of Table 4, and the “Non-uniform 1.4x” configuration in Table 5. In Table 6, higher accuracies result from applying sparsity only to specific modules rather than across the entire model.
>
> > Context-Output Aware Sparsity
>
> Lines 263–264 describe our strategy as “selectively preserving dense computation for output tokens while applying sparsity to the context,” meaning only a portion of output tokens are processed densely, as reflected in Figure 8. We will revise the text to avoid potential ambiguity.
>
> > Table 5 Speedup Columns
>
> The “1.4x Speedup” and “1.6x Speedup” columns indicate runtime targets by adjusting sparsity ratios. Both uniform and non-uniform configurations are calibrated to these targets for fair comparison. The 1.4x (1/0.71) non-uniform setting, with 79.8% accuracy on Math10K, directly corresponds to the SparseLoRA LLaMA3-8B result in Table 2; the 1.6x setting explores a more aggressive sparsity trade-off.
>
> > "Sequence averaging" in the SVD estimator
>
> “Sequence averaging” reduces sparsity estimation cost by averaging activations over the sequence dimension for SVD inputs to compute a single channel score. This is particularly effective for FFN layers, where token-level differences are less critical, and it minimizes overhead while maintaining performance (see Table 3).
>
> > FLOPs and Runtime Optimizations in Tables 1-2.
>
> The reductions reported in Tables 1 and 2 stem from applying contextual channel sparsity guided by our SVD estimator to expensive linear layers. Sequence averaging further minimizes the overhead of sparsity estimation, making the cost of identifying sparse channels negligible.
>
> > Trained to Convergence and Uniformity of Sparsity Ratios
>
> No additional training was performed during the sensitivity analysis. As shown in Figure 7, we apply sparsity to the pretrained LLaMA2-7B model at individual layers (with others kept dense) and evaluate its performance on CSR170K. This analysis identifies robust sparsity patterns that are then uniformly applied across all models and tasks.
>
> > Comparison with Ma et al. (2024)
>
> Ma et al.’s method differs from ours in many ways: it uses structured pruning with full fine-tuning, resulting in full-model memory usage. In contrast, SparseLoRA integrates structured pruning with LoRA adapters, reducing both computation and memory. Moreover, their importance metrics (Wanda, MaxiP) involve element-wise operations that incur high overhead (e.g., on A6000 GPUs), nearly canceling speedup gains when we experimented Wanda for Table 6. Our SVD-based sparsity estimator relies on efficient matrix multiplications with low-rank weights, ensuring minimal overhead across GPUs.
>
> > Performance drop not negligible
>
> SparseLoRA offers a flexible speed–accuracy trade-off. For applications where maximum accuracy is crucial, LoRA remains viable; for efficiency-critical scenarios, SparseLoRA delivers substantial speedups with minimal accuracy loss. Here we provide an updated, more conservative sparsity configuration for LLaMA2-13B and LLaMA3-8B that slightly increase FLOPs (0.59 → 0.62 for LLaMA2-13B and 0.60 → 0.63 for LLaMA3-8B) but yield nearly identical runtime (0.74 → 0.76 for LLaMA2-13B and 0.71 for LLaMA3-8B) and match or improve accuracy:
>
> | |#FLOPs|Runtime|Avg|GSM8K|SVAMP|MAWPS|
> |---|:-----:|:-----:|:-----:|:-----:|:-----:|:-----:|
> |LLaMA2-13B+LoRA|1|1|63.3%|50.7%|59.0%|80.4%|
> |LLaMA2-13B+SparseLoRA|0.62|0.76|63.5%|49.3%|58.7%|82.6%|
> |LLaMA3-8B+LoRA|1.00|1.00|80.0%|71.1%|79.5%|89.5%|
> |LLaMA3-8B+SparseLoRA|0.63|0.71|80.0%|70.9%|79.4%|89.9%|
>
> These results show that, with proper configuration, SparseLoRA can achieve comparable or better accuracy than LoRA while delivering strong runtime benefits.

---

> > ### Comment · Reviewer_as16 · 2025-04-06
> >
> > I thank the authors for their rebuttal. While some of my concerns have been addressed, my main concern still remains:
> >
> > > Practical application
> >
> > I am still unconvinced about the practicality of the method. The method involves a number of overheads over LoRA. Arguably, LoRA is much simpler and very practical to run. With efficiency tricks like gradient checkpointing and quantization (for e.g., QLoRA, LoftQ), is accessible even on consumer grade GPUs. Moreover, if the sparsity ratios need to be adjusted to obtain equivalent performance, SparseLoRA needs to be ablated over a range of sparsity ratios for tasks from different domains. This is important because if the starting point for completely novel tasks needs to be a low sparsity, using LoRA would already yield performance close to the maximum achievable performance.

---

> > > ### Author Response · Authors · 2025-04-08
> > >
> > > We thank the reviewer for the thoughtful comments and for raising important points regarding the practicality of our method.
> > >
> > > We emphasize that SparseLoRA is not intended as a replacement for LoRA but rather as a complementary enhancement. Techniques like gradient checkpointing and quantization (e.g., QLoRA, LoftQ) primarily target memory savings but often often come at the cost of increased runtime (as illustrated in Figure 1 of our paper). These methods are therefore orthogonal to our approach. In fact, SparseLoRA can be combined with techniques like QLoRA to simultaneously benefit from reduced memory consumption and improved runtime efficiency — a direction we believe could be highly valuable in practice – as shown below:
> > >
> > > ### LLaMA3-8B Commonsense170K
> > > | Method | Runtime | Mean | BoolQ | PIQA | Social-IQA | HellaSwag | Winogrande | ARC-Easy | ARC-Challenge | OpenBookQA |
> > > |--------|---------|------|-------|------|------------|-----------|------------|----------|---------------|------------|
> > > | QLoRA | 1.00 | 87.2% | 74.4% | 89.4% | 83.3% | 95.4% | 89.2% | 93.3% | 84.2% | 88.7% |
> > > | +SparseLoRA | 0.76 | 87.1% | 74.8% | 89.6% | 83.1% | 95.3% | 88.6% | 93.2% | 83.7% | 89.0% |
> > >
> > > ### LLaMA3-8B Math10K
> > > | Method | Runtime | Mean | GSM8K | SVAMP | MAWPS |
> > > |--------|---------|------|-------|-------|-------|
> > > | QLoRA | 1.00 | 80.8% | 71.4% | 80.2% | 90.3% |
> > > | +SparseLoRA | 0.74 | 80.5% | 71.3% | 79.9% | 90.8% |
> > >
> > > Regarding sparsity configurations and their generalizability, we clarify that **our sparsity settings are primarily model-dependent rather than task-dependent**. As discussed in our response to Reviewer pTrd, the sparsity ratios remain largely consistent across tasks, with minor adjustments primarily made to achieve specific runtime targets rather than to maintain accuracy. We further validated this generality by applying the same near-lossless sparsity configurations on LLaMA-3 across diverse benchmarks—including Commonsense Reasoning, Math10K, and GLUE (also provided in our response to Reviewer pTrd)—achieving consistent strong performance without extensive per-task tuning, addressing the reviewer's concern about practical applicability in new settings.
> > >
> > > Lastly, we note that SparseLoRA provides a comparable level of **simplicity and plug-and-play applicability** to methods like QLoRA and LoftQ. Both of these existing methods leverage quantization, with LoftQ additionally requiring mixed-precision setups to achieve optimal trade-offs—indicating that efficient PEFT approaches naturally carry some complexity. SparseLoRA integrates easily within this framework, making it equally practical for real-world adoption. Our initial QLoRA integration results above, obtained without extensive tuning (a "speed run"), already demonstrate promising efficiency gains and near-lossless accuracy. Further optimization or targeted tuning would likely yield even stronger results in practice.
> > >
> > > Thus, SparseLoRA presents a valuable and practical enhancement for existing PEFT methods, aligning closely with the community's ongoing efforts toward efficient yet accurate model adaptation.

---

### Decision · Program_Chairs · 2025-05-01

**Decision:**

Accept (poster)

**Comment:**

This paper received 4 weak accept. After careful consideration of the submission, the reviews and rebuttal, this meta-review recommends *weak accept*.

On the positive aspects, the proposed structural sparsity is a relevant contribution in improving computational efficiency of PEFT (LoRA) methods. The paper also presents comprehensive engineering of applying the proposed structural sparsity in terms of per-layer sensitivity analysis, context/output-aware sparsity, and progressive sparsity scheduling, with promising empirical results obtained. The rebuttal did a good job in addressing reviewers' concerns, leading to some upgraded recommendations.

There are several concerns remained: (i) Whether SparseLoRA’s complexity outweighs its benefits compared to simpler PEFT methods like QLoRA, especially for users not constrained by compute. The authors argued SparseLoRA is complementary and showed it can integrate with QLoRA. (ii) While SparseLoRA matches or slightly outperforms LoRA in low-FLOP regimes, it doesn’t necessarily improve peak accuracy. (iii) The writing and some definitions in the paper should be improved, which the authors promised to revise in the rebuttal.